# FreeTraj: Tuning-Free Trajectory Control via Noise Guided Video Diffusion

## Abstract

Diffusion model has demonstrated remarkable capability in video generation, which further sparks interest in introducing trajectory control into the generation process. While existing works mainly focus on training-based methods (*e.g.*, conditional adapter), we argue that diffusion model itself allows decent control over the generated content without requiring any training. In this study, we introduce a tuning-free framework to achieve trajectory-controllable video generation, by imposing guidance on both noise construction and attention computation. Specifically, **1)** we first show several instructive phenomena and analyze how initial noises influence the motion trajectory of generated content. **2)** Subsequently, we propose **FreeTraj**, a tuning-free approach that enables trajectory control by modifying noise sampling and attention mechanisms. **3)** Furthermore, we extend FreeTraj to facilitate longer and larger video generation with controllable trajectories. Equipped with these designs, users have the flexibility to provide trajectories manually or opt for trajectories automatically generated by the LLM trajectory planner. Extensive experiments validate the efficacy of our approach in enhancing the trajectory controllability of video diffusion models. Generated video samples are available at the anonymous website: `https://FreeTraj.github.io`.

## 1 Introduction

Thanks to the powerful modeling capabilities of diffusion models, significant progress has been made in open-world visual content generation, as evidenced by numerous foundational text-to-video models (Wang et al., 2023b; Chen et al., 2024b). These models can generate vivid dynamic content based on arbitrary text prompts. However, while text prompts offer flexibility, they fall short of concretely expressing users' intentions, particularly regarding geometric control. Although existing trajectory control works primarily rely on training ControlNet-like structures (Wang et al., 2023c; Chen et al., 2023d), we contend that diffusion model itself contains the potential of substantial control over the generated content without necessitating additional training. In this paper, we aim to investigate the dynamics modeling mechanisms of video diffusion models and explore the possibility of explicitly controlling object trajectories by leveraging their internal properties. While most of the existing efforts are made by modifying text embeddings or adjusting attention mechanisms to enable control or editing (Ren et al., 2024; Geyer et al., 2023), the influence of initial noises on video motion remains under-explored.

For text-to-video diffusion models, there is considerable diversity in the generated content (*e.g.*, motion trajectories) from the same text prompt, depending on the choice of initial noises. This phenomenon motivates us to raise a question: Is it possible to regulate the motion trajectories with some designs over initial noises? FreeInit (Wu et al., 2023c) has observed that low-frequency signals are more resistant to additive noises, which makes the diffusion model biased to inherit layout or shape information from the initial noises. Consequently, by arranging the low-frequency components of noises across frames, we can manipulate the inter-frame content correlation, *i.e.,* the temporal movements of the generated video. However, this constraint is not that reliable because the inter-frame region correlation is not directly aligned with object semantics. Prior works (Jain et al., 2023a; Ma et al., 2023a) have demonstrated that trajectories can also be influenced by adjusting the attention weights assigned to different objects in some specific areas. Thus, to achieve object-level-based trajectory control, we propose to utilize text-based attention to locate the target objects in cooperation with noise space manipulation.

However, introducing alterations to the noise or attention mechanism carries the risk of causing artifacts in the generated videos. For example, applying a local mask to the self-attention operation can cause partially abnormal values because this diverges from the case encountered by the models during training. Furthermore, these minor anomalies can propagate through subsequent layers and become amplified in the following denoising steps, ultimately filling the target region with artifacts. We call such a phenomenon as *attention isolation*. Previous work (Jain et al., 2023a) suffers from this problem and is easy to generate artifacts in the areas with masks. In our proposed **FreeTraj** system, we are fully aware of this issue and mitigate these risks by applying our operations to the noise and attention mechanisms with a tailor-made scheme. Instead of hard attention masks used in Peekaboo (Jain et al., 2023a), our designed soft attention masks relieve the phenomenon of attention isolation. This approach strikes a balance between staying close to the training distribution and maintaining the ability to control trajectories.

In addition, FreeTraj can be seamlessly integrated into the long video generation framework, enriching the motion trajectories within the generated long videos. Current video generation models are typically trained on a restricted number of frames, leading to limitations in generating high-fidelity long videos during inference. FreeNoise (Qiu et al., 2023) proposes a tuning-free and time-efficient paradigm for longer video generation based on pre-trained video diffusion models. Although FreeNoise brings satisfactory video quality and visual consistency, it has no guarantee for the various trajectories of generated objects, which are supposed to appear in long videos. With the help of some technical points proposed by FreeNoise, our FreeTraj successfully generates trajectory-controllable long videos. FreeTraj is also valuable in larger video generation. When we directly generate videos with resolutions larger than those in the model training process, we will easily get results with duplicated main objects (He et al., 2024). However, FreeTraj will constrain the information of the main objects to the target areas. Signals of main objects are suppressed in other areas thus the duplication phenomenon will be reduced.

Our contributions are summarized as follows: **1)** We investigate the mechanism of how initial noises influence the trajectory of generated objects through several instructive phenomenons. **2)** We propose **FreeTraj**, an effective paradigm for tuning-free trajectory control with both noise guidance and attention guidance. **3)** We extend the control mechanism to achieve longer and larger video generation with a controllable trajectory.

## 2 RELATED WORK

**Diffusion Models for Visual Generation.** Diffusion models have revolutionized image and video generation, showcasing their ability to produce high-quality samples. DDPM (Ho et al., 2020) and Guided Diffusion (Dhariwal & Nichol, 2021) are groundbreaking works that show diffusion models can generate high-quality samples. To improve efficiency, LDM (Rombach et al., 2022) introduces latent space diffusion models that operate in a lower-dimensional space, reducing computational costs and training time, which serves as the foundation of Stable Diffusion. SDXL (Podell et al., 2023) builds upon Stable Diffusion, achieving high-resolution image generation. Pixart-alpha (Chen et al., 2023b) replaces the backbone with a pure transformer, resulting in high-quality and cost-effective image generation. In terms of video generation, VDM (Ho et al., 2022b) is the first video generation model that utilizes diffusion. LVDM (He et al., 2022) takes it a step further by proposing a latent video diffusion model and hierarchical LVDM framework and achieves very long video generation. Align-Your-Latents (Blattmann et al., 2023b) and AnimateDiff (Guo et al., 2023) propose to insert temporal transformers into pre-trained text-to-image generation models to achieve text-to-video (T2V) generation. VideoComposer (Wang et al., 2023c) presents a controllable text-to-video generation framework that is capable of controlling both spatial and temporal signals. VideoCrafter (Chen et al., 2023a; 2024b) and SVD (Blattmann et al., 2023a) scale up the latent video diffusion model to large datasets. Lumiere (Bar-Tal et al., 2024) introduces temporal downsampling to the space-time U-Net. Sora (OpenAI, 2024) is a closed-source video generator that has impressive results announced most recently and has garnered much attention. In this work, we choose VideoCrafter 2.0 (referred to as VideoCrafter in the rest of the paper) as our pre-trained base model, as it is a current state-of-the-art open-sourcing model based on the comprehensive evaluations from Vbench (Huang et al., 2023b) and EvalCrafter (Liu et al., 2023b).

**Trajectory Control in Video Generation.** Given the critical role of motion in video generation, research on motion control in generated videos has garnered increasing attention. One intuitive method involves utilizing motion extracted from reference videos (Liu et al., 2023a; Wei et al., 2023; Zhao et al., 2023a; Li et al., 2023; Chen et al., 2024a; Yatim et al., 2024). For instance, approaches such as Tune-A-Video (Wu et al., 2023a), MotionDirector (Zhao et al., 2023b), and LAMP (Wu et al., 2023b) use specific videos as references to generalize their motions to various generated videos. Although these methods achieve significant motion control in video generation, they require training for each reference motion. To circumvent the need for specific motion training, ControlNet-like structures, such as VideoComposer (Wang et al., 2023c) and Control-A-Video (Chen et al., 2023d), employ depths, sketches, or moving vectors extracted from reference videos as conditions to control the motion of generated videos. However, these methods are limited to generating videos with pre-existing motions, constraining their creativity and customization. In contrast, controlling the motion of generated videos using trajectories or bounding boxes offers more flexibility and user-friendliness (Chen et al., 2023c; Deng et al., 2023; Wang et al., 2024; Yang et al., 2024; Huang et al., 2023a). While training-based methods (Chen et al., 2023c; Yin et al., 2023a; Deng et al., 2023; Wang et al., 2023d; 2024) have demonstrated significant motion controllability, they demand substantial computing resources and are labor-intensive during data collection. Consequently, inspired by previous work applying attention mask for image editing (Hertz et al., 2022; Cao et al., 2023), several training-free trajectory control approaches (Yang et al., 2024; Huang et al., 2023a) have emerged. These methods, such as Peekaboo (Jain et al., 2023b) and TrailBlazer (Ma et al., 2023b), employ explicit attention control to direct the movement of generated objects according to specified trajectories. Our work also adopts a training-free approach. We enhance motion controllability in generated videos by imposing guidance on both noise construction and attention computation, resulting in improved performance in both motion control and video quality.

## 3 METHODOLOGY

### 3.1 PRELIMINARIES: VIDEO DIFFUSION MODELS

Video Diffusion Models (VDM) (Ho et al., 2022a) denotes diffusion models used for video generation, which formulates a fixed forward diffusion process to gradually add noise to the 4D video data $\boldsymbol{x}_0 \sim p(\boldsymbol{x}_0)$ and learn a denoising model to reverse this process. The forward process contains $T$ timesteps, which gradually add noise to the data sample $\boldsymbol{x}_0$ to yield $\boldsymbol{x}_t$ through a parameterization trick:

$$q(\boldsymbol{x}_t|\boldsymbol{x}_{t-1}) = \mathcal{N}(\boldsymbol{x}_t; \sqrt{1-\beta_t}\boldsymbol{x}_{t-1}, \beta_t\boldsymbol{I}), \qquad q(\boldsymbol{x}_t|\boldsymbol{x}_0) = \mathcal{N}(\boldsymbol{x}_t; \sqrt{\bar{\alpha}_t}\boldsymbol{x}_0, (1-\bar{\alpha}_t)\boldsymbol{I}), \quad (1)$$

where $\beta_t$ is a predefined variance schedule, $t$ is the timestep, $\bar{\alpha}_t = \prod_{i=1}^{t} \alpha_i$, and $\alpha_t = 1 - \beta_t$. The reverse denoising process obtains less noisy data $\boldsymbol{x}_{t-1}$ from the noisy input $\boldsymbol{x}_t$ at each timestep:

$$p_\theta(\boldsymbol{x}_{t-1} \mid \boldsymbol{x}_t) = \mathcal{N}(\boldsymbol{x}_{t-1}; \boldsymbol{\mu}_\theta(\boldsymbol{x}_t, t), \boldsymbol{\Sigma}_\theta(\boldsymbol{x}_t, t)). \quad (2)$$

Here $\boldsymbol{\mu}_\theta$ and $\boldsymbol{\Sigma}_\theta$ are determined through a noise prediction network $\boldsymbol{\epsilon}_\theta(\boldsymbol{x}_t, t)$, which is supervised by the following objective function, where $\boldsymbol{\epsilon}$ is sampled ground truth noise and $\theta$ is the learnable network parameters.

$$\min_\theta \mathbb{E}_{t, \boldsymbol{x}_0, \boldsymbol{\epsilon}} \|\boldsymbol{\epsilon} - \boldsymbol{\epsilon}_\theta(\boldsymbol{x}_t, t)\|_2^2, \quad (3)$$

Once the model is trained, we can synthesize a data point $\boldsymbol{x}_0$ from random noise $\boldsymbol{x}_T$ by sampling $\boldsymbol{x}_t$ iteratively. Considering the high complexity and inter-frame redundancy of videos, Latent Diffusion Model (LDM) (Rombach et al., 2022) is widely adopted to formulate the diffusion and denoising process in a more compact latent space. Latent Video Diffusion Models (LVDM) is realized through perceptual compression with a Variational Auto-Encoder (VAE) Kingma & Welling (2014), where an encoder $\mathcal{E}$ maps $\boldsymbol{x}_0 \in \mathbb{R}^{3 \times F \times H \times W}$ to its latent code $\boldsymbol{z}_0 \in \mathbb{R}^{4 \times F \times H' \times W'}$ and a decoder $\mathcal{D}$ reconstructs the video $\boldsymbol{x}_0$ from the $\boldsymbol{z}_0$. Then, the diffusion model $\theta$ operates on the video latent variables to predict the noise $\hat{\boldsymbol{\epsilon}}$.

$$\boldsymbol{z}_0 = \mathcal{E}(\boldsymbol{x}_0), \quad \hat{\boldsymbol{x}}_0 = \mathcal{D}(\boldsymbol{z}_0) \approx \boldsymbol{x}_0, \quad \hat{\boldsymbol{\epsilon}} = \boldsymbol{\epsilon}_\theta(\boldsymbol{z}_t, \boldsymbol{y}, t), \quad (4)$$

where $\boldsymbol{y}$ denotes conditions like text prompts. Most mainstream LVDMs (Blattmann et al., 2023b; Wang et al., 2023b; Chen et al., 2023a) are implemented by a UNet equipped with convolutional

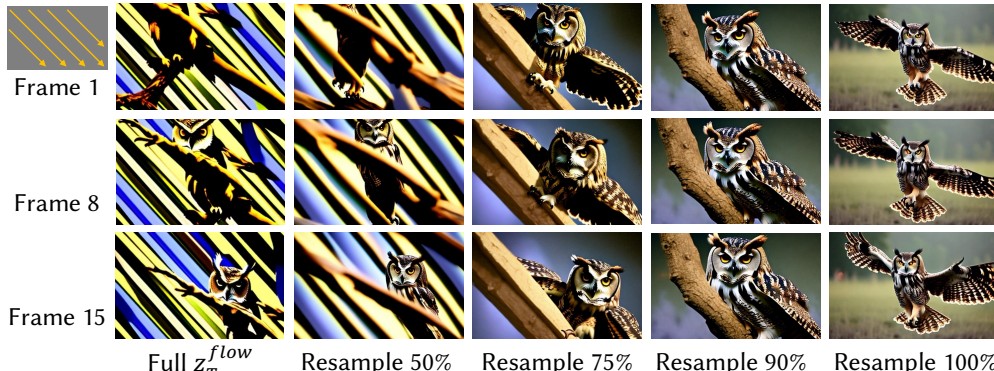

Figure 1: **Noise resampling of initial high-frequency components.** Gradually increasing the proportion of resampled high-frequency information in the frame-wise shared noises can significantly reduce the artifact in the generated video. However, this also leads to a gradual loss in trajectory control ability. A resampling percentage of 75% strikes a better balance between maintaining control and improving the quality of the generated video.

modules, spatial attentions, and temporal attentions. The basic computation block (whose feature input and output are $\mathbf{h}$ and $\mathbf{h}'$ respectively) could be denoted as:

$$\mathbf{h}' = \text{TT}(\text{ST}(\text{Tconv}(\text{Conv}(\mathbf{h}, t)), \mathbf{y})), \quad \text{TT} = \text{Proj}_{\text{in}} \circ (\text{Attn}_{\text{temp}} \circ \text{Attn}_{\text{temp}} \circ \text{MLP}) \circ \text{Proj}_{\text{out}}. \tag{5}$$

Here Conv and ST are residual convolutional block and spatial transformer, while Tconv denotes temporal convolutional block and TT denotes temporal transformers, serving as cross-frame operation modules.

## 3.2 Noise Influence on Trajectory Control

During the training process of the video diffusion model, it cannot fully corrupt the semantics when adding noise, leaving substantial spatio-temporal correlations in the low-frequency components (Wu et al., 2023c). Those low-frequency correlations may still contain information about trajectory. Therefore, if we simulate the noises of the training process and manually add some spatio-temporal correlations in the low-frequency components, we have a chance to control the trajectory of the generated video.

**Noise Flow.** Our first attempt is to make the noise flow among frames. Instead of randomly sampling initial noises for all frames, we only sample the noise for the first frame. Then we move the noise from the top-left to the bottom-right with stride 2 and repeat this operation until we get initial noises $z_T^{flow}$ for all frames. Specially, initial noise $\epsilon$ for each frame $f$ in position $[i, j]$ is:

$$\epsilon[i, j]^f = \epsilon[(i-2) \pmod{H}, (j-2) \pmod{W}]^{f-1}. \tag{6}$$

After denoising $z_T^{flow}$, although we will get a video with strong artifacts (Figure 1), we can still find a valuable phenomenon: objects and textures in the video also flow in the same direction (top-left to bottom-right). This phenomenon verifies that the trajectory of the initial noises can guide the motion trajectory of generated results.

**High-Frequency Noise Resampling.** Artifacts in noise flow are mainly caused by deviation from the independent random distribution of the initial noises. Therefore, if we resample some new random independent noises to replace some dependent noises in $z_T^{flow}$, more realistic results are expected to be generated. According to the analysis of FreeInit (Wu et al., 2023c), the trajectory information is mainly obtained in the low-frequency noise. Therefore, we use Fourier Transformation to resample high-frequency noise and get new latent $\tilde{z}_T$ to perform further denoising:

$$\begin{aligned} \mathcal{F}_{z_T}^{low} &= \mathcal{FFT}_{3D}(z_T) \odot \mathcal{H}, \\ \mathcal{F}_{\eta}^{high} &= \mathcal{FFT}_{3D}(\eta) \odot (1 - \mathcal{H}), \\ \tilde{z}_T &= \mathcal{IFFT}_{3D}\left(\mathcal{F}_{z_T}^{low} + \mathcal{F}_{\eta}^{high}\right), \end{aligned} \tag{7}$$

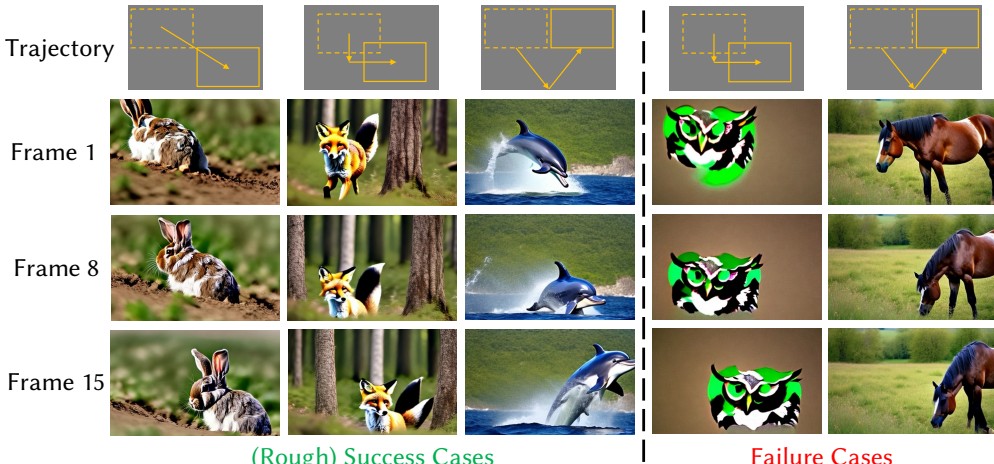

Figure 2: **Trajectory control via frame-wise shared low-frequency noise.** The success cases on the left demonstrate that the moving objects in the generated videos can be roughly controlled by sharing low-frequent noise across the bounding boxes of the given trajectory. However, the precision of control and the success rate remain somewhat constrained, as evidenced by the failure instances on the right.

where $\mathcal{FFT}_{3D}$ is the Fast Fourier Transformation operated on both spatial and temporal dimensions, and $\mathcal{IFFT}_{3D}$ is the Inverse Fast Fourier Transformation that maps noise back from the blended frequency domain. $\mathcal{H}$ is the spatial-temporal Low Pass Filter (LPF), which is a tensor of the same shape as the latent. $\eta$ is a newly sampled random noise to replace the high-frequency of the original noise. In this case, $z_T = z_T^{flow}$.

Figure 1 shows that the visual quality is significantly improved as the proportion of high-frequency noise resampled increases. Correspondingly, the flow phenomenon is weakened. When $90\%$ high-frequency noise is resampled, the flow is almost stopped with only some similar textures remaining (*e.g.,* branches from top-left to bottom right). Overall, $75\%$ resampling strikes a good balance between sportiness and image quality.

**Trajectory Injection.** In noise flow, all objects in the foreground and background tend to move toward the direction of flow. If we only control the flow happening in the local area with some trajectories, can we guide the only main object to move following the corresponding trajectories? To answer it, we design some trajectories from simple to complex and make the flow area occupy a quarter of the area, as shown in the first row of Figure 2.

Instead of directly denoising random noises, we inject trajectory into the initial noises. We first initialize a random local noise $\epsilon_{local}$ according to the area of the input mask and $F$ frames of random noises $[\epsilon_1, \epsilon_2, ..., \epsilon_F]$ independently. Then for each frame $f$, the initial noise $\epsilon_f$ will be replaced by the $\epsilon_{local}$ if in the area of the input mask:

$$\tilde{\epsilon}_f[i,j] = \begin{cases} \epsilon_f[i,j] & \text{if } M_f[i,j] = 0 \\ \epsilon_{local}[i^*,j^*] & \text{if } M_f[i,j] = 1 \end{cases}, \tag{8}$$

where $\epsilon_f, \epsilon_{local} \sim \mathcal{N}(\mathbf{0}, \mathbf{I})$, $M_f$ is the input mask of frame $f$, and $M_f[i,j] = 1$ if the position $(i,j)$ is inside the bounding box of trajectory. $M_f[i,j] = 0$ otherwise. $(i^*, j^*)$ is the corresponding local position in the box.

As shown in the left of Figure 2, some objects are well generated and follow the trajectory injected in initial noises although they may not be fully aligned with the given bounding boxes. While these objects move along the trajectory, they will also try to follow the prior knowledge of the physical world contained in the model (*e.g.* dolphins cannot go too far from the sea after jumping). And the right of Figure 2 shows some failure cases. They are either poor in visual quality or in trajectory alignment.

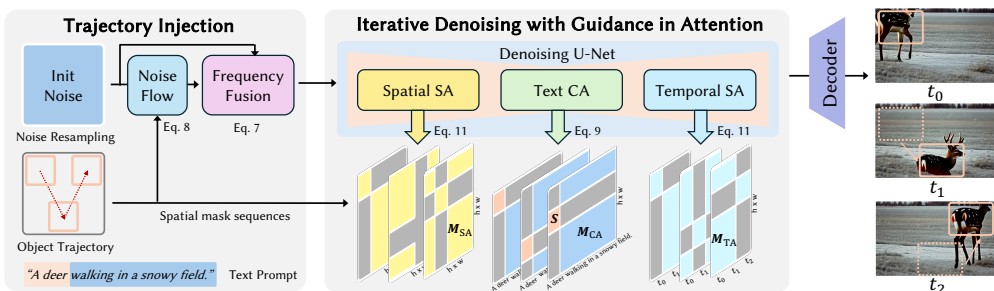

Figure 3: **An overview of FreeTraj.** Our framework mainly contains two parts: guidance in noise and guidance in attention. For noise, we inject the target trajectory into the low-frequency part. For attention, we design different reweighing strategies according to the supposed behaviors in different attention layers. Here $\mathcal{S}$, $M_{CA}$, $M_{SA}$, and $M_{TA}$ are different attention masks.

Based on those observations, although we can utilize initial noises to guide the trajectory, we still need to involve additional control mechanisms to achieve accurate trajectory control, especially when the target trajectory deviates from a prior knowledge of the physical world contained in the model.

### 3.3 THE FRAMEWORK OF FREETRAJ

Given a target bounding box for a foreground object in the video, we suppose the pre-trained video model to generate results whose trajectory is aligned with the given box. To achieve that, we propose **FreeTraj**, which designs guidance in both noise and attention as shown in Figure 3.

#### 3.3.1 GUIDANCE IN NOISE

As analyzed in Section 3.2, frame-wise shared low-frequency noise can guide the trajectory of generated objects. Therefore, we inject trajectory in the initial noises through Equation 8. To reduce the phenomena of attention isolation, we still need to remove some of the injected noises through High-Frequency Noise Resampling (Equation 7).

#### 3.3.2 GUIDANCE IN ATTENTION

Object trajectories in generated videos with only noise guidance still tend to follow the prior information of the video model. In addition, the controlled object will be automatically selected by the model according to the training data distribution and cannot be manually specified. To make the control more precise and assignable, we also add trajectory guidance in attention. There are three kinds of attention layers in the UNet of VideoCrafter (Chen et al., 2024b): spatial cross-attention, spatial self-attention, and temporal self-attention. Unlike previous work Peekaboo (Jain et al., 2023a) directly masks the foreground and background respectively for all attention layers, we design different strategies according to the supposed behaviors in different attention layers. All attention editing is performed in the early steps $t \in \{T, \ldots, T - N\}$ of the denoising process, where $T$ is the total number of denoising timesteps, and $N$ is the number of timesteps for attention editing.

**Attention Isolation.** We find the previous designs in attention may cause attention isolation. It is a phenomenon that some regions become isolated either spatially or temporally and rarely pay attention to information outside their own region. This is often caused by the values in this area deviating too much from the training distribution. Unlucky, it is difficult for this region to restore itself to normal levels through valuable information from the other regions due to the isolation. Therefore, it is necessary to avoid attention isolation when we modify the attention mechanism without re-training. We will discuss more in the ablation study and appendix.

**Cross Attention Guidance.** Spatial cross-attention is the only place for prompts to inject the information from text embedding. Originally, the model would assign the object according to the prompts and initial noises. It is a random and unpredictable behavior. To force the model to only generate the target object in the given bounding box, we first add guidance to the cross-attention. Given query $Q$, key $K$, value $V$ of cross-attention, and the re-scaled binary 2D attention masks $M_a$ and $M'_a$, which indicate the foreground and background areas of the generated video respectively.

Our guided cross-attention is:

$$\text{GuidedCrossAttention}(Q, K, V, M_a, M'_a) = (\text{softmax}\left(\frac{QK^T}{\sqrt{d}} + \mathcal{M}\right) + \mathcal{S})V,$$

$$\text{where } \mathcal{S}[i,j] = \begin{cases} 0 & \text{if } M_a[i,j] = 0 \\ \alpha\, g(i,j) & \text{if } M_a[i,j] = 1 \end{cases}, \text{and } \mathcal{M}[i,j] = \begin{cases} -\infty & \text{if } M'_a[i,j] = 0 \\ 0 & \text{if } M'_a[i,j] = 1 \end{cases}.$$

(9)

Here $\alpha$ is a coefficient to enhance the influence of target prompts in the foreground and $g(\cdot, \cdot)$ is a Gaussian weight (Ma et al., 2023a). Note that the attention masks $M_a, M'_a \in \{0,1\}^{d_q \times d_k}$, where $d_q$ and $d_k$ are the lengths of queries and keys, respectively. They are attained with a given prompt $P$ and the target mask $M^f_{\text{target}}[i]$ of frame $f$ ($M^f_{\text{target}}[i]$ is a 1-D flatten form of $M_f$ in Eq. 8). In the cross-attention layer, $M_a$ and $M'_a$ are respectively denoted as $M_{CA}$ and $M'_{CA}$, where

$$M^f_{CA}[i,j] = \text{fg}\left(M^f_{\text{target}}[i]\right) * \text{fg}(P[j]),$$

$$M'^f_{CA}[i,j] = \left(1 - \text{fg}\left(M^f_{\text{target}}[i]\right)\right) * (1 - \text{fg}(P[j])),$$

(10)

where fg is a function that takes a pixel or a text token as input, returning 1 if it corresponds to the foreground of the video, and 0 otherwise.

**Self Attention Guidance.** Self-attention consists of the spatial part and temporal part. Without mandatory constraints, the information in the foreground and background will interact. In this case, the video model may still generate target objects at unexpected locations. Therefore, we design guided self-attention:

$$\text{GuidedSelfAttention}(Q, K, V, M_a) = \text{softmax}\left(\frac{QK^T}{\sqrt{d}} \odot \mathcal{W}\right)V,$$

$$\text{where } \mathcal{W}[i,j] = \begin{cases} \beta & \text{if } M_a[i,j] = 0 \\ 1 & \text{if } M_a[i,j] = 1 \end{cases}.$$

(11)

Here $\beta$ is a coefficient to weaken the influence of the interaction of foreground and background. Compared to the hard mask using $-\infty$ to forbid the interaction of foreground and background, this soft mask design can avoid some artifacts caused by attention isolation.

The attention mask $M_a$ designed in self-attention follows the Peekaboo (Jain et al., 2023a). Specifically, in the spatial self-attention layer, $M_a$ is denoted as $M_{SA}$, where

$$M^f_{SA}[i,j] = \text{fg}\left(M^f_{\text{target}}[i]\right) * \text{fg}\left(M^f_{\text{target}}[j]\right)$$

$$+ \left(1 - \text{fg}\left(M^f_{\text{target}}[i]\right)\right) * \left(1 - \text{fg}\left(M^f_{\text{target}}[j]\right)\right),$$

(12)

and in the temporal self-attention layer, $M_a$ is denoted as $M_{TA}$, where

$$M^i_{TA}[f,k] = \text{fg}\left(M^f_{\text{target}}[i]\right) * \text{fg}\left(M^k_{\text{target}}[i]\right)$$

$$+ \left(1 - \text{fg}\left(M^f_{\text{target}}[i]\right)\right) * \left(1 - \text{fg}\left(M^k_{\text{target}}[i]\right)\right).$$

(13)

## 3.4 Longer Video Generation

FreeTraj can also be seamlessly integrated into the longer video generation framework FreeNoise (Qiu et al., 2023) to generate rich motion trajectories in long videos. FreeNoise mainly applies Local Window Fusion to the temporal attention to guarantee visual quality and utilize Noise Rescheduling in the noise initialization to reserve video consistency.

Local Window Fusion divides the temporal attention into several overlapped local windows along the temporal dimension and then fuses them together. In order to cooperate with Local Window Fusion, our guidance in temporal attention is only applied within each Local Window Fusion. Noise Rescheduling reuses and shuffles the sub-fragment of initial noises. To avoid our guidance in noise being destroyed, Equation 8 and Equation 7 are applied after Noise Rescheduling. Through the combined new framework, our method can achieve trajectory control over a long video sequence without any fine-tuning (Figure 7 in the appendix).

Table 1: **Quantitative comparison of trajectory control.** FreeTraj achieves competitive performance in metrics about video quality and gains the best scores in metrics that are related to trajectory control.

| Method | FVD ($\downarrow$) | KVD ($\downarrow$) | CLIP-SIM ($\uparrow$) | mIoU ($\uparrow$) | CD ($\downarrow$) |
|---|---|---|---|---|---|
| Direct | **118.19** | **-2.28** | **0.980** | 0.161 | 0.225 |
| MotionCtrl (Wang et al., 2023d) | 825.80 | 68.07 | 0.939 | – | 0.248 |
| MotionCtrl-256 (Wang et al., 2023d) | 601.35 | 47.60 | 0.938 | – | 0.245 |
| Peekaboo (Jain et al., 2023a) | 403.00 | 25.30 | 0.963 | 0.235 | 0.179 |
| TrailBlazer (Ma et al., 2023a) | 556.00 | 42.14 | 0.958 | 0.179 | 0.219 |
| Ours | 436.22 | 29.85 | 0.956 | **0.281** | **0.154** |
| Ours-ShortMove | 369.22 | 21.00 | 0.971 | 0.344 | 0.119 |

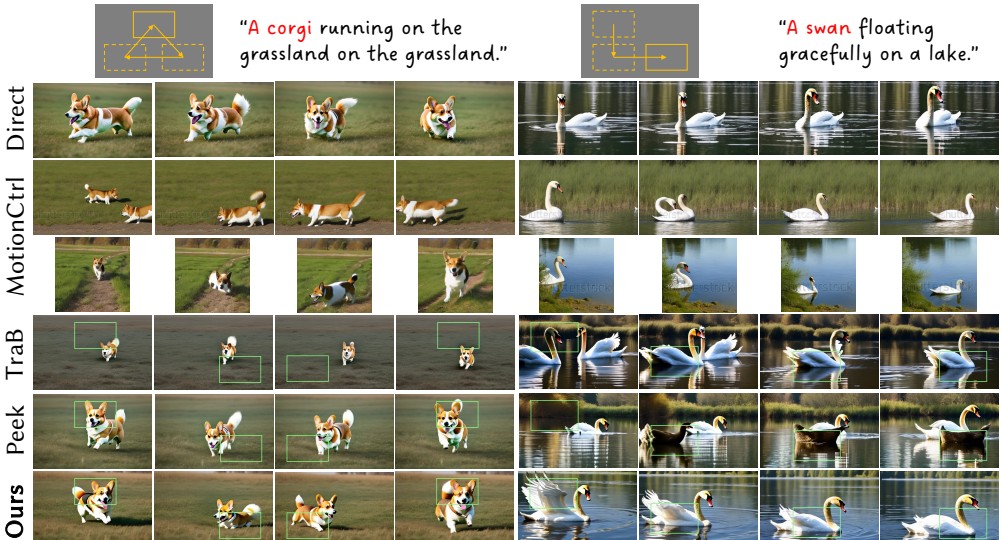

Figure 4: **Qualitative comparison of trajectory control.** We compare our proposed FreeTraj with direct inference (Direct), Peekaboo (Peek), and TrailBlazer (TraB). FreeTraj successfully generates high-fidelity results and is more accurate for trajectory control.

## 4 EXPERIMENTS

Based on performance and accessibility considerations, we choose the recently published open-source video diffusion model, VideoCrafter (Chen et al., 2024b), as our pre-trained video model in this paper. All experiments are conducted based on this model. The inference resolution is fixed at $320 \times 512$ pixels and the video length is 16 frames unless stated otherwise.

**Evaluation Metrics.** To evaluate video quality, we report Fréchet Video Distance (FVD) (Unterthiner et al., 2018), Kernel Video Distance (KVD) (Unterthiner et al., 2019). Since the tuning-free methods are supposed to keep the quality of the original pre-trained inference, we calculate the FVD and KVD between the original generated videos and videos generated by trajectory control methods. We use CLIP Similarity (CLIP-SIM) (Radford et al., 2021) to measure the semantic similarity among frames. In addition, we utilize the off-the-shelf detection model, OWL-ViT-large (Minderer et al., 2022), to obtain the bounding box of the synthesized objects. Then Mean Intersection of Union (mIoU) and Centroid Distance (CD) are calculated to evaluate the trajectory alignment. CD is the distance between the centroid of the generated object and the input mask, normalized to 1. When OWL-ViT-large fails to detect the target object in the generated videos, the farthest point will be assigned as the penalty in CD.

### 4.1 EVALUATION OF TRAJECTORY CONTROL

We first directly sample videos using the pre-trained model without trajectory control as a base reference. Then we compare our proposed FreeTraj to other trajectory-controllable video generation methods with diffusion models, MotionCtrl (Wang et al., 2023d), Peekaboo (Jain et al., 2023a) and

Table 2: **User study.** Users are requested to pick the best one among our proposed FreeTraj with the other baseline methods in terms of trajectory alignment, video-text alignment, and video quality.

| Method | Trajectory Alignment | Video-Text Alignment | Video Quality |
|---|---|---|---|
| Peekaboo (Jain et al., 2023a) | 6.48% | 15.12% | 13.58% |
| TrailBlazer (Ma et al., 2023a) | 8.03% | 6.79% | 6.79% |
| Ours w/o Noise | 19.75% | 15.43% | 15.74% |
| Ours | **65.74%** | **62.65%** | **63.89%** |

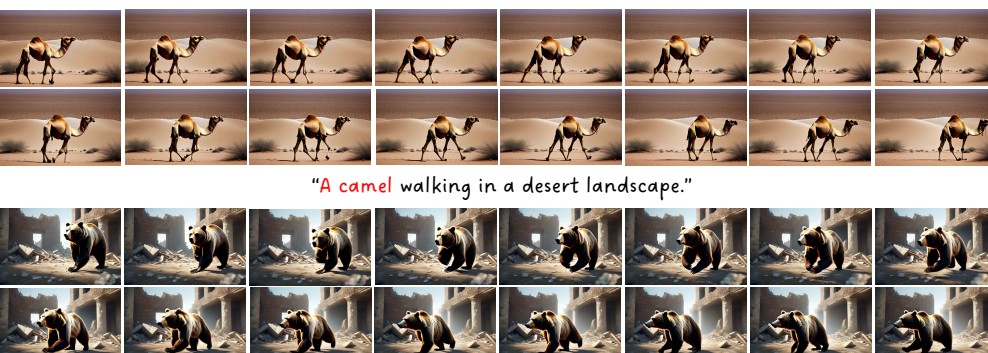

"A camel walking in a desert landscape."

"A bear running in the ruins, photorealistic, 4k, high definition."

Figure 5: **Qualitative results with short movements.** As shown in all generated 16 frames, the video quality and motion coherence are well preserved when we only require FreeTraj to generate some results with short movements (like from left to right or reverse).

TrailBlazer (Ma et al., 2023a). MotionCtrl achieves trajectory control with the trainable object motion control module, requiring re-training for each base model. To support the input format of MotionCtrl, we convert the sequence of bounding boxes to the sequence of central points and omit the mIoU. In addition, MotionCtrl only releases one version for trajectory control which is trained on $256 \times 256$ resolution. However, the resolution of our baseline is $320 \times 512$. For a fair comparison, we report the results of both resolutions. Peekaboo and TrailBlazer are two tuning-free methods that control trajectory through masked attention.

As shown in the first line of Figure 4, the original pre-trained VideoCrafter tends to generate objects that act around the center of the frame with limited movements. MotionCtrl can control the trajectory of objects but does not force the object center to align with the trajectory accurately, thus gaining a poor CD score. Notably, the object motion control module of MotionCtrl lacks transferability thus obtains worse performance in $320 \times 512$. In addition, the frame quality of MotionCtrl is obviously inferior to that of other methods because it is trained based on VideoCrafter1 while can not be integrated into VideoCrafter2 directly. TrailBlazer is weak in control because it only applies the control in spatial cross-attention and temporal self-attention while information will leak through spatial self-attention. Videos generated by Peekaboo follow the given trajectory controls better because all kinds of attention are masked without information leakage. However, Peekaboo generates an additional black swan with weird artifacts, which is probably caused by the hard attention mask used in self-attention layers. Our FreeTraj controls trajectory via the combined effect of initial noise and attention layers, thus succeeding in driving the target object following the given trajectories with vivid motions. Our method also achieves competitive scores in FVD, KVD, and CLIP-SIM, which exhibits the reliable video quality generated by our method.

**User Study.** Furthermore, we conducted a user study to evaluate our results based on human subjective perception. Participants were asked to watch the generated videos from all methods, with each example displayed in a random order to avoid bias. They were then instructed to select the best video in three evaluation aspects: trajectory alignment, video-text alignment, and video quality. The results, as shown in Table 2, demonstrate that our approach outperforms the baseline methods by a significant margin, achieving the highest scores in all aspects. Notably, our method received nearly 70% votes in terms of trajectory alignment. This user study confirms the superiority of our approach in terms of trajectory alignment, video-text alignment, and video quality.

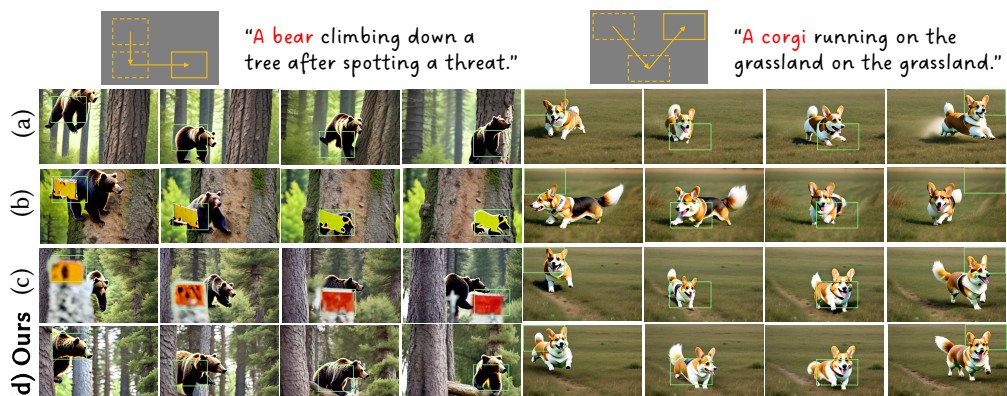

Figure 6: **Ablation results.** (a) No noise guidance, (b) no high-frequency noise resampling, (c) hard attention mask, and (d) our whole method.

**Movement Scale.** We evaluate control abilities on some complex trajectories (*e.g.,* top-left → bottom-left → bottom-right) within 16 frames. However, even for a real video, presenting such a long-range movement within 16 frames may lead to either motion incoherence or motion blur. Therefore, we also add some examples with shorter movements (*e.g.,* left → right), whose range of movement is close to the setting of previous works. As shown in Figure 5, the video quality and motion coherence are well preserved. In addition, short movements also bring better FVD and KVD because this behavior is similar to reference videos directly generated by VideoCrafter.

### 4.2 ABLATION STUDIES

**Ablation of Noise Guidance.** To show the effectiveness of noise guidance, we run our designed attention guidance solely. Figure 6 (a) shows that pure attention guidance can also control the trajectory but may lose some accuracy.

**Ablation of Attention Isolation.** We also study two settings that may cause attention isolation. The first one uses no high-frequency noise resampling when applying trajectory injection in initial noises (Equation 7). The second one employs the hard attention mask in Equation 11. Usually, diffusion models have some robustness to deal with the input with small deviation and recover it to generate qualified results. However, both of these two strategies will easily cause the value of the attention layer to deviate far from the data distribution in the training stage. It will lead to attention isolation where isolated regions almost pay no attention to other regions, losing the chance to recover back to the normal distribution. As shown in Figure 6 (b) and (c), blocky artifacts appear and follow the given trajectory in the generated videos. In addition, those artifacts happen to fall at the position of the attention mask or inject local noise.

### 5 CONCLUSION

In conclusion, our study has revealed several instructive phenomenons about how initial noises influence the generated results of video diffusion models. Leveraging the noise guidance and combining it with careful modifications to the attention mechanism, we introduce a tuning-free framework, **FreeTraj**, for trajectory-controllable video generation using diffusion models. We demonstrate that diffusion models inherently possess the capability to control generated content without additional training. By guiding noise construction and attention computation, we enable trajectory control and extend it to longer and larger video generation. Although not shown in this paper, our approach offers flexibility for users to provide trajectories manually or automatically generated by the LLM trajectory planner. Extensive experiments validate the effectiveness of our approach in enhancing the trajectory controllability of video diffusion models, providing a practical and efficient solution for generating videos with desired motion trajectories. However, this tuning-free paradigm is still limited by the underlying model, such as the consistency of object appearance that easily changes during large movements. We hope the study of initial noises can also inspire the training strategy of basic video models.

## 6 ETHICS STATEMENT

The primary objective of this project is to empower individuals without specialized expertise to create video art more effectively. Our paradigm, based on the pre-trained video diffusion model, assists the model in generating trajectory-controllable videos. It is important to note that the content generated by our tuning-free paradigm remains rooted in the original model. As a result, regulators only need to oversee the original video generation model to ensure adherence to ethical standards, and our algorithm does not introduce any additional ethical concerns.

## 7 REPRODUCIBILITY STATEMENT

We have introduced the algorithm and implementation details in detail in the paper. A researcher familiar with the video diffusion model should be able to reproduce our method. In addition, we will release our code after acceptance for better promotion.

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

## A    APPENDIX: IMPLEMENTATION

### A.1    HYPERPARAMETERS

During sampling, we perform DDIM sampling (Song et al., 2020) with 50 denoising steps, setting DDIM eta to 0. The inference resolution is fixed at $320 \times 512$ pixels and the video length is 16 frames in the normal setting. The video length of longer inference is 64 frames and the inference resolution of larger inference is $640 \times 512$ pixels. The scale of the classifier-free guidance is set to 12. $\alpha$ in Equation 9 is $\frac{0.25}{len\_target\_prompts \times proportion\_target\_box}$ and $\beta$ in Equation 11 is 0.01. The kernel division in Equation 9 is 3.0 and the kernel shape is the same as the mask shape.

For quantitative comparison, we generate a total of 896 videos for each inference method, utilizing 56 prompts. We initialize 16 random initial noises for each prompt for direct inference. For trajectory control methods, each prompt is applied to 8 different trajectories with 2 random initial noises. The height and width of the trajectory bounding box are randomly chosen as 0.3, 0.35, or 0.4 of the canvas size.

In the user study, we mixed our generated videos with those generated by the other three baselines. A total of 27 users were asked to pick the best one according to the trajectory alignment, video-text alignment, and video quality, respectively.

### A.2    PROMPTS

Our prompts are mostly extended from previous baselines(Jain et al., 2023a; Ma et al., 2023a) but replace some prompts that conflict with object movement, like standing or lying.

- **A woodpecker** climbing up a tree trunk.
- **A squirrel** descending a tree after gathering nuts.
- **A bird** diving towards the water to catch fish.
- **A frog** leaping up to catch a fly.
- **A parrot** flying upwards towards the treetops.
- **A squirrel** jumping from one tree to another.
- **A rabbit** burrowing downwards into its warren.
- **A satellite** orbiting Earth in outer space.
- **A skateboarder** performing tricks at a skate park.
- **A leaf** falling gently from a tree.
- **A paper plane** gliding in the air.
- **A bear** climbing down a tree after spotting a threat.
- **A duck** diving underwater in search of food.
- **A kangaroo** hopping down a gentle slope.
- **An owl** swooping down on its prey during the night.
- **A hot air balloon** drifting across a clear sky.
- **A red double-decker bus** moving through London streets.
- **A jet plane** flying high in the sky.
- **A helicopter** hovering above a cityscape.
- **A roller coaster** looping in an amusement park.
- **A streetcar** trundling down tracks in a historic district.
- **A rocket** launching into space from a launchpad.

- **A deer** walking in a snowy field.
- **A horse** grazing in a meadow.
- **A fox** running in a forest clearing.
- **A swan** floating gracefully on a lake.
- **A panda** walking and munching bamboo in a bamboo forest.
- **A penguin** walking on an iceberg.
- **A lion** walking in the savanna grass.
- **An owl** flying in a tree at night.
- **A dolphin** just breaking the ocean surface.
- **A camel** walking in a desert landscape.
- **A kangaroo** jumping in the Australian outback.
- **A colorful hot air balloon** tethered to the ground.
- **A corgi** running on the grassland on the grassland.
- **A corgi** running on the grassland in the snow.
- **A man** in gray clothes running in the summer.
- **A knight** riding a horse on a race course.
- **A horse** galloping on a street.
- **A lion** running on the grasslands.
- **A dog** running across the garden, photorealistic, 4k.
- **A tiger** walking in the forest, photorealistic, 4k, high definition.
- **Iron Man** surfing on the sea.
- **A tiger** running in the forest, photorealistic, 4k, high definition.
- **A horse** running, photorealistic, 4k, volumetric lighting unreal engine.
- **A panda** surfing in the universe.
- **A chihuahua** in an astronaut suit floating in the universe, cinematic lighting, glow effect.
- **An astronaut** waving his hands on the moon.
- **A horse** galloping through a meadow.
- **A bear** running in the ruins, photorealistic, 4k, high definition.
- **A barrel** floating in a river.
- **A dark knight** riding a horse on the grassland.
- **A wooden boat** moving on the sea.
- **A red car** turning around on a countryside road, photorealistic, 4k.
- **A majestic eagle** soaring high above the treetops, surveying its territory.
- **A bald eagle** flying in the blue sky.

# B APPENDIX: LONGER AND LARGER VIDEO GENERATION

## B.1 RELATED WORK OF LONG VIDEO GENERATION.

Long video generation is a challenging but important problem in video generation. TATs (Ge et al., 2022), longvideoGAN (Ge et al., 2022), LVDM (He et al., 2022), and flexible diffusion (Harvey et al., 2022) achieve long video generation in small domains and without textual guidance. Phenaki (Villegas et al., 2022), NUWA-Infinity (Liang et al., 2022), NUWA-XL (Yin et al., 2023b), and Sora (OpenAI, 2024) are text-guided long video generation approaches for open-domain generation. Animate-A-Story (He et al., 2023) achieves multi-scene long video generation via character consistency. Streamingt2v (Henschel et al., 2024) and FlexiFilm (Ouyang et al., 2024) are training-based methods that train a conditional module on top of pre-trained video diffusion models conditioning on previous

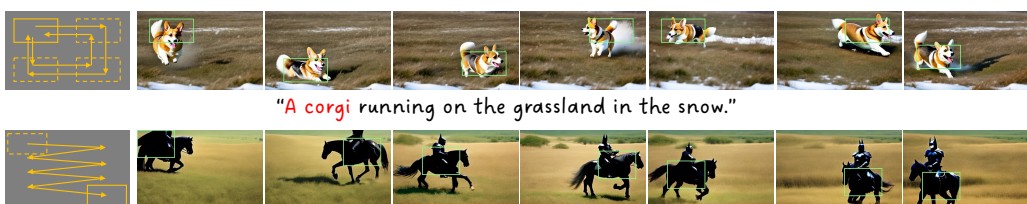

Figure 7: **Longer video generation.** Longer video generation allows us to plan some complex trajectories. FreeTraj succeeds in generating rich motion trajectories in long videos.

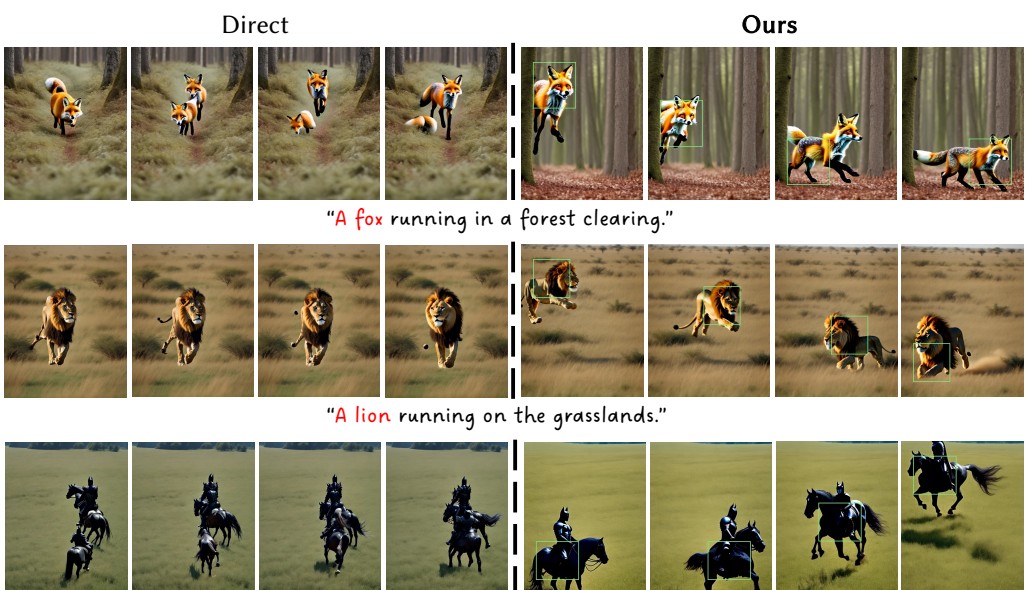

Figure 8: **Larger video generation.** Directly generating larger videos will easily lead to the results with duplicated main objects anywhere. FreeTraj plans the trajectory for the main object and suppresses the duplication phenomenon.

frames. Genlvideo (Wang et al., 2023a) and FreeNoise (Qiu et al., 2023) are recently proposed tuning-free methods for generating longer videos based on pre-trained video diffusion models to extend their generated length. In this work, we propose a tuning-free approach for long video generation based on long-term trajectory control.

### B.2 RESULTS OF LONGER GENERATION

FreeTraj can be integrated into the longer video generation framework FreeNoise (Qiu et al., 2023). With the help of some technical points proposed by FreeNoise, our FreeTraj successfully generated trajectory-controllable long videos. As shown in Figure 7, we plan two complex paths and FreeTraj succeeds in generating rich motion trajectories in long videos.

### B.3 RESULTS OF LARGER GENERATION

When we directly use pre-trained video diffusion models to generate videos with higher resolutions compared to those in training, they will easily generate results with duplicated main objects anywhere He et al. (2024). However, FreeTraj will plan the trajectory for the main object, and information

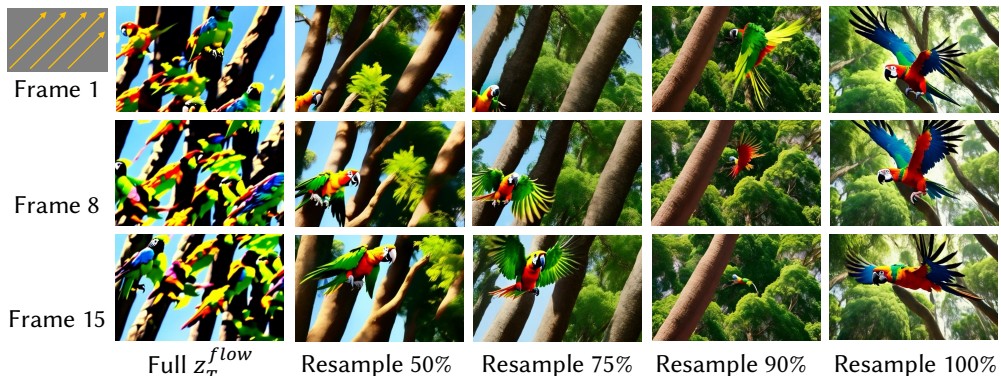

Figure 9: **Noise resampling of initial high-frequency components.** Gradually increasing the proportion of resampled high-frequency information in the frame-wise shared noises can significantly reduce the artifact in the generated video. However, this also leads to a gradual loss in trajectory control ability. A resampling percentage of 75% strikes a better balance between maintaining control and improving the quality of the generated video.

about the main object will be reduced out of the target areas. Therefore, the duplication phenomenon will be suppressed by FreeTraj (Figure 8).

## C  APPENDIX: MORE OBSERVATIONS

### C.1  MORE ABOUT NOISE FLOW

Here we show another direction of noise flow. Instead of randomly sampling initial noises for all frames, we only sample the noise for the first frame. Then we move the noise from the bottom-left to the top-right with stride 2 and repeat this operation until we get initial noises $z_T^{flow}$ for all frames. Specially, initial noise $\epsilon$ for each frame $f$ in position $[i, j]$ is:

$$\epsilon[i, j]^f = \epsilon[(i + 2) \pmod{H}, (j - 2) \pmod{W}]^{f-1}. \tag{14}$$

After denoising $z_T^{flow}$, results in Figure 9 show that objects and textures in the video also flow in the same direction (bottom-left to top-right). This phenomenon verifies that the trajectory of the initial noises can have an impact on the motion trajectory of the generated result. When the proportion of high-frequency noise resampled increases, the visual quality is significantly improved. Correspondingly, the flow phenomenon is weakened.

The initial noise guidance also works for some other similar base video models. As shown in Figure 10, the Noise Flow phenomenon still holds on AnimateDiff (Guo et al., 2023).

### C.2  ATTENTION ISOLATION IN TEMPORAL DIMENSION

Usually, we initialize 16 frames of random noises independently. Instead of normal sampling, we try partial repeated sampling by partially repeating some initial noises:

$$\begin{aligned}
&\text{Normal Sampling: } [\epsilon_1, \epsilon_2, \epsilon_3, \epsilon_4, \epsilon_5, \epsilon_6, \epsilon_7, \epsilon_8, \epsilon_9, \epsilon_{10}, \epsilon_{11}, \epsilon_{12}, \epsilon_{13}, \epsilon_{14}, \epsilon_{15}, \epsilon_{16}], \\
&\text{Partial Repeated Sampling: } [\boldsymbol{\epsilon_1}, \boldsymbol{\epsilon_1}, \boldsymbol{\epsilon_1}, \boldsymbol{\epsilon_1}, \epsilon_2, \epsilon_3, \epsilon_4, \epsilon_5, \epsilon_6, \epsilon_7, \epsilon_8, \epsilon_9, \boldsymbol{\epsilon_{10}}, \boldsymbol{\epsilon_{10}}, \boldsymbol{\epsilon_{10}}, \boldsymbol{\epsilon_{10}}].
\end{aligned} \tag{15}$$

Since spatio-temporal correlations in the low-frequency components of initial noises will guide the trajectory of generated objects, partial repeated sampling for initial noises will bring typical motion mode. As shown in Figure 11 (b), the owl is stationary in the beginning and ending frames and only has significant action in the middle frames. However, due to the attention isolation, frames of generated results have obvious artifacts. We visualize one heat map of temporal attention and find that stationary frames mainly pay attention to frames with the same initial noises. When calculating the attention weights received by isolated frames, manually splitting a portion of attention weights from isolated frames to other frames will remove artifacts. As shown in Figure 11 (c), an owl is well generated and its motion still fits the mode in (b).

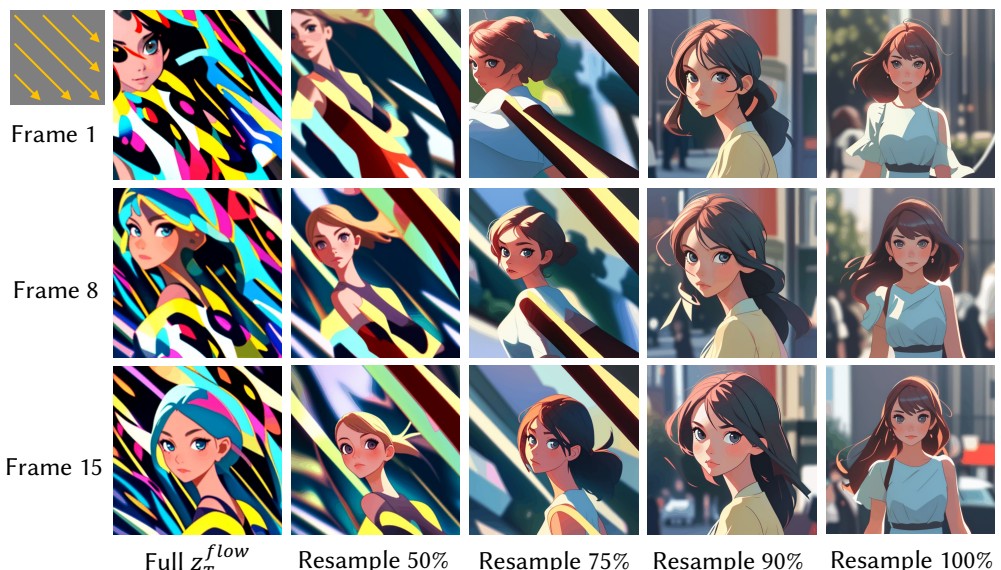

Figure 10: **Noise flow in AnimateDiff.** In AnimateDiff, objects and textures in the video also flow in the same direction as the initial noises.

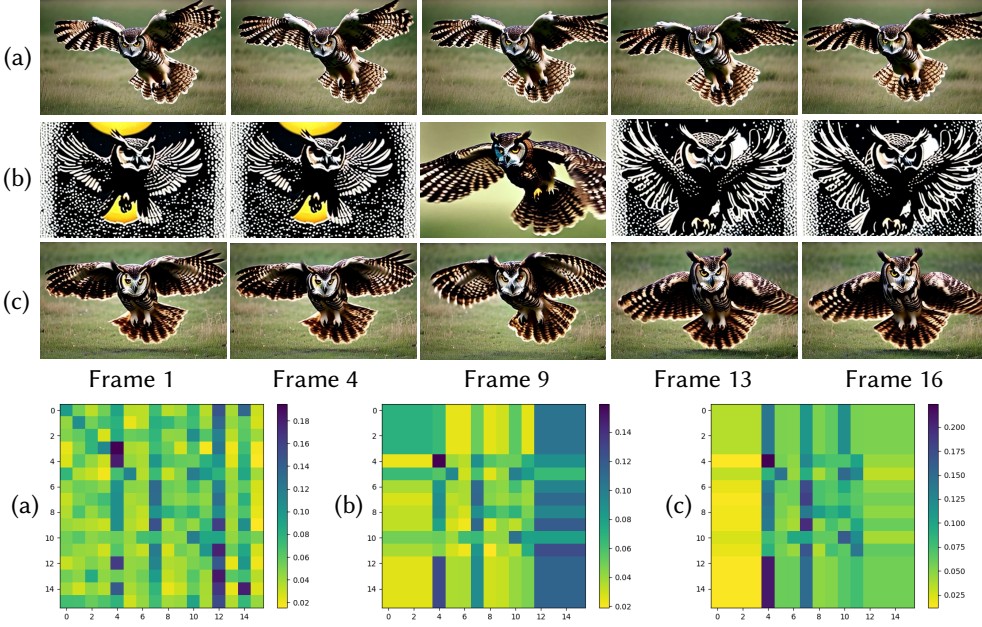

Figure 11: **Attention isolation in temporal dimension.** Compared to normal sampling for initial noises (a), partial repeated sampling will lead to significant attention isolation in the temporal dimension and bring strong artifacts (b). When calculating the attention weights received by isolated frames, manually splitting a portion of attention weights from isolated frames to other frames will remove artifacts (c).

Table 3: **Quantitative comparison of ablations.** Dynamics is the score of dynamic degree (Huang et al., 2023b). The best results are marked in **bold**, and the second best results are marked by underline.

| Method | FVD ($\downarrow$) | KVD ($\downarrow$) | CLIP-SIM ($\uparrow$) | mIoU ($\uparrow$) | CD ($\downarrow$) | Dynamics ($\uparrow$) |
|---|---|---|---|---|---|---|
| No Noise Guidance | **390.65** | **24.48** | **0.964** | 0.277 | 0.156 | 0.973 |
| No Noise Resampling | 513.79 | 39.88 | 0.960 | 0.279 | 0.167 | 0.973 |
| Higher Intensity Control | 697.72 | 62.06 | 0.951 | **0.322** | **0.149** | **1.000** |
| Ours | 436.22 | 29.85 | 0.956 | 0.281 | 0.154 | 0.982 |

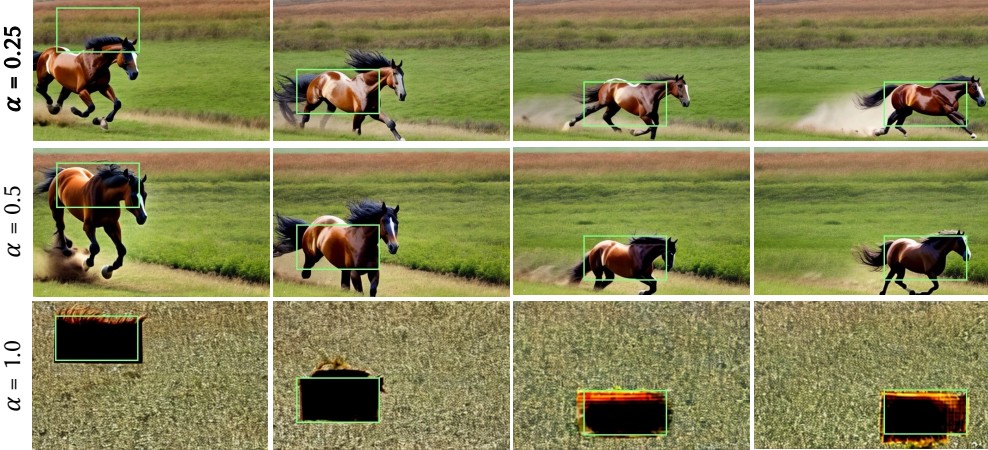

"A horse galloping through a meadow."

Figure 12: **Visualization of control intensity.** When increasing the control intensity, the generated objects will follow the given bounding boxes more closely. However, meaningless patterns will be generated when the intensity is large.

### C.3 QUANTITATIVE ABLATION

We also conduct the ablation study quantitatively. For the setting of higher intensity control, we increase the $\alpha$ in Equation 9 from $0.25$ to $0.5$. As shown in Table 3, our final setting achieves a competitive performance in both video quality and trajectory control.

### C.4 CONTROL INTENSITY

We can easy adjust the control intensity by modifying the $\alpha$ in Equation 9. In this paper, $\alpha = 0.25$ is a default value to guarantee that most generated cases do not contain artifacts. However, as shown in Figure 12, $\alpha = 0.5$ is a better choice for higher control intensity. Users can get a better traject-controllable result by sampling more times with different random seeds. Meaningless patterns will be generated when the intensity is large.

### C.5 METHOD COMPATIBILITY

We test FreeTraj on another diffusion-based video generation method, AnimateDiff. As shown in Figure 13, FreeTraj effectively achieves trajectory control in AnimateDiff, potentially making it a versatile tool in video synthesis, especially for applications requiring rapid deployment without extensive training data.

### C.6 LLM-PLANED GENERATION

We slightly modify the prompt from the previous work (Lian et al., 2023) and the LLM will plan the bounding boxes for each frame. The results are shown in Figure 14.

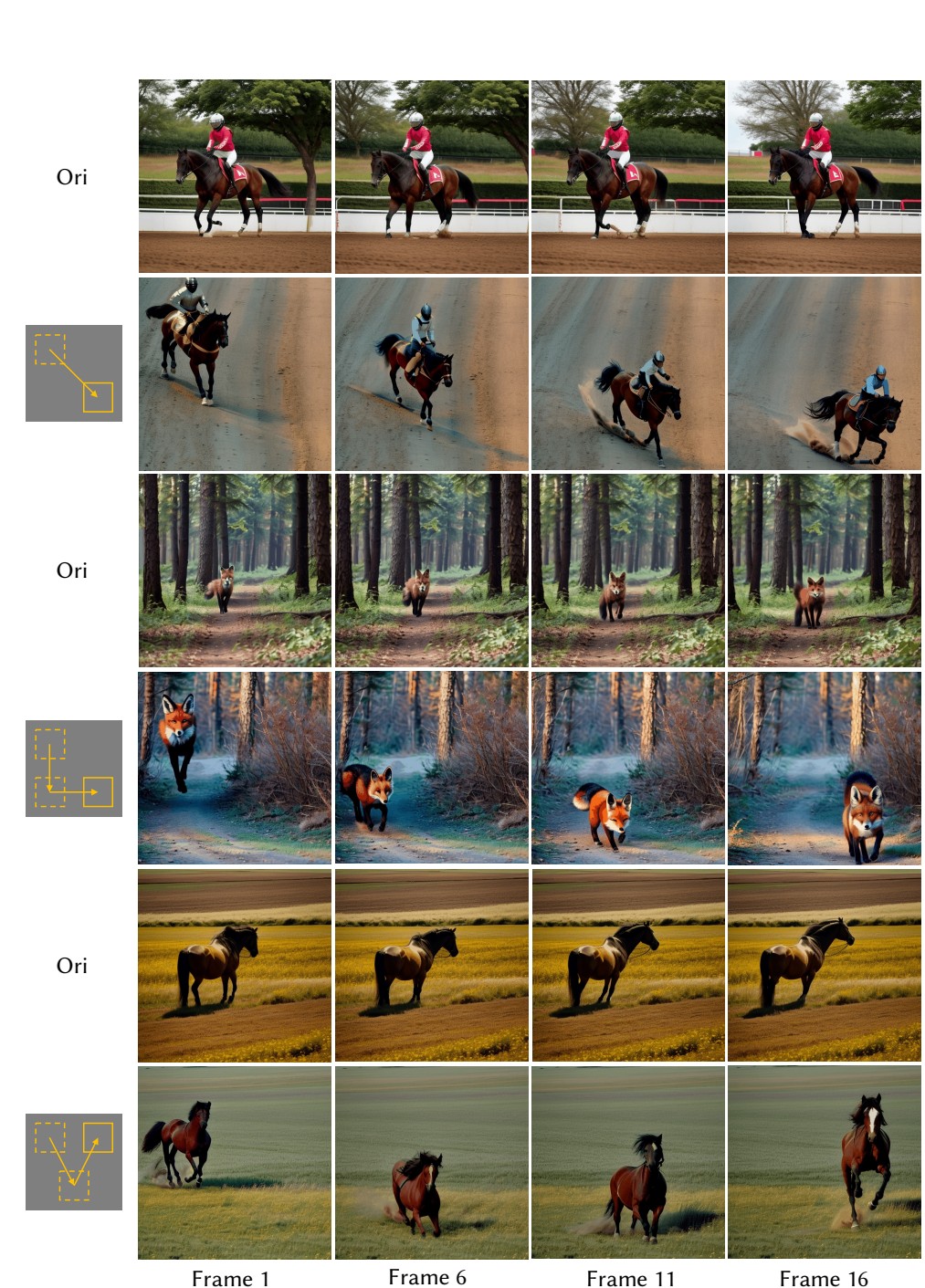

Frame 1       Frame 6       Frame 11       Frame 16

Figure 13: **Trajectory control in AnimateDiff.** FreeTraj is also compatible with AnimateDiff.

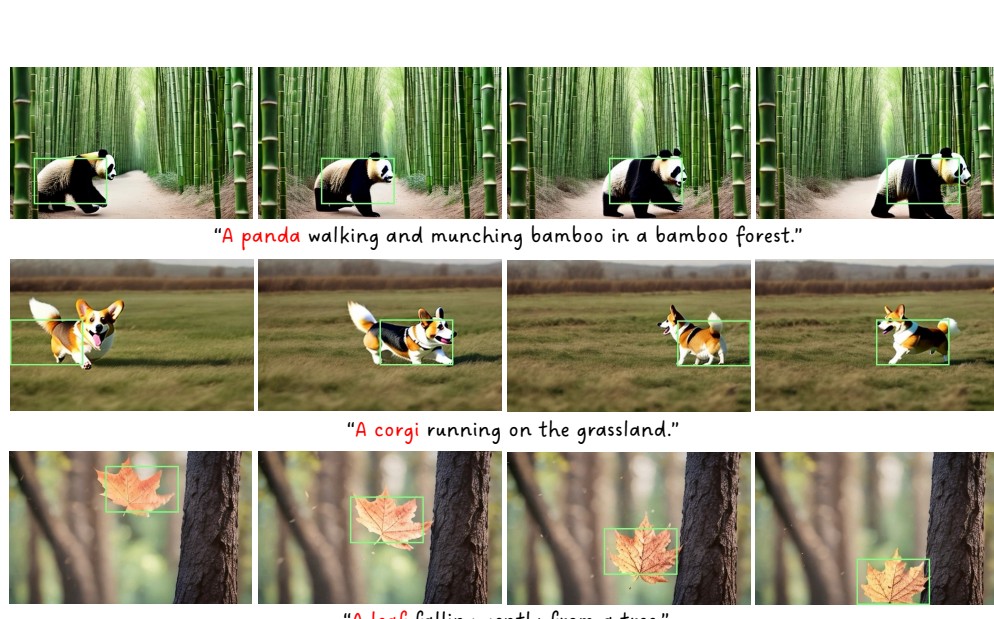

Figure 14: **LLM-planed generation.** Users can use the LLM to plan the trajectory as the input of FreeTraj.