# OpenReview forum: "FreeTraj: Tuning-Free Trajectory Control via Noise Guided Video Diffusion"
_ICLR.cc/2025/Conference — Submitted to ICLR 2025_

### Official Review · Reviewer_2SGJ · 2024-10-21

**Soundness:** 3
**Presentation:** 3
**Contribution:** 3
**Rating:** 6
**Confidence:** 4

**Summary:**

This paper introduces FreeTraj, a tuning-free framework for trajectory-controllable video generation using diffusion models, eliminating the need for additional training. It controls motion trajectories by guiding noise sampling and attention mechanisms. FreeTraj supports both manual and LLM-generated trajectories and enables longer video generation with precise trajectory control. Extensive experiments validate its effectiveness in enhancing trajectory controllability in video diffusion models.

**Strengths:**

1. The method is training free unlike most existing works.
2. Extensive quantitative evaluation.
3. Methods are intuitive and the paper is easy to follow.
4. Superiority over baselines compared.
5. Providing observation before they build their method sounds convincing and well-structured.
6. Applications of Longer / Larger video generation.

**Weaknesses:**

1. By "LLM trajectory planner", do the authors mean like LLM-grounded video diffusion [1]. Plus, I think LLM-grounded video diffusion would serve as a good baseline to be compared (ignoring the llm planning part). Also, if the LLM trajectory planner is not implemented or showcased, the authors should not sell this point.

2. Does the method increase computation burden? Providing and comparing information on the memory / time consumption would increase value of the paper.

3. How long and big are the "long" and "large" video generation in the results of appendix?

4. I think the intuition behind High-Frequency Noise Resampling is well explored in both image/video diffusion works. Not just FreeInit, but also in Diffusion-Motion-Transfer [2] (their initial noise preparation stage). And the novelty is limited.

5. It's hard to grasp what attention isolation is easily. Providing visualizations in relevant sections would help reviewers understand the issue and how they overcome it.

6. Is Cross Attention Guidance method not explored in diffusion-based image generation/editing literature?

7. Isn't TrackDiffusion [3] a relevant work? If so, how does the model perform compared to the work?

8. Quantitative ablations studies would add value of the paper.

[1] LLM-grounded Video Diffusion Models, ICLR 2024

[2] Space-Time Diffusion Features for Zero-Shot Text-Driven Motion Transfer, CVPR 2024

[3] TrackDiffusion: Tracklet-Conditioned Video Generation via Diffusion Models, Arxiv 2023

**Questions:**

Please refer to the weakness section.

---

> ### Author Response · Authors · 2024-11-25
>
> Thanks for your valuable comments. We summarize and answer your questions below.
>
> > **Q1: If the LLM trajectory planner is not implemented or showcased, the authors should not sell this point.**
>
> Thanks for pointing it out. We slightly modified the prompt from the paper, LLM-Grounded Video Diffusion Models, and have showcased the results in Figure 14 of the revised version. In addition, we agree that LLM-grounded video diffusion would serve as a good baseline. However, the LLM may give some weird paths and still need human selection manually, posing some hindrances to scaling operations. This problem may be solved in the future with the development of LLMs.
>
> > **Q2: Does the method increase computation burden?**
>
> Thanks for pointing it out. On a single NVIDIA V100 GPU, the inference time of the direct generation is 106.28s, while the unoptimized FreeTraj is 155.04s, bringing 45.88% additional time. If we only calculate all the attention masks once and store them for each time step, FreeTraj will bring only around 10% extra time.
>
> > **Q3: How long and big are the "long" and "large" video generation in the results of appendix?**
>
> Sorry for the confusion. Due to page limitation, we discuss it in the appendix of the original submission. This information is listed in the implementation part of the original submission: “The inference resolution is fixed at 320x512 pixels and the video length is 16 frames in the normal setting. The video length of longer inference is 64 frames and the inference resolution of larger inference is 640x512 pixels.”
>
> To make it easier for the supplementary material to be indexed, we have added an overview for each part of the appendix in the revised version.
>
> > **Q4: The intuition behind High-Frequency Noise Resampling is well explored in both image/video diffusion works.**
>
> Thanks for pointing it out. One of our core novelties is exactly trajectory injection, initializing the noise without any DDIM inversion or re-nosing, which is never achieved in FreeInit or other previous work. Diffusion-Motion-Transfer [1] is a relevant work, and we have added it to the Related Work.
>
> > **Q5: It's hard to grasp what attention isolation is easily.**
>
> Sorry for the confusion. Due to page limitation, we discuss it in the appendix of the original submission: Figure 11: Compared to normal sampling for initial noises (a), partial repeated sampling will lead to significant attention isolation in the temporal dimension and bring strong artifacts (b). When calculating the attention weights received by isolated frames, manually splitting a portion of attention weights from isolated frames to other frames will remove artifacts (c).
>
> Attention isolation in the spatial dimension is similar to that in the temporal dimension (it is more easily to visualize the temporal attention map). I hope this visualization will help to grasp attention isolation.
>
> To make it easier for the supplementary material to be indexed we have added an overview for each part of the appendix in the revised version.
>
> > **Q6: Is Cross Attention Guidance method not explored in diffusion-based image generation/editing literature?**
>
> Thanks for your advice. We have added some related papers [2, 3] in the revised version. If we still miss some important literature, please let us know.
>
> > **Q7: Isn't TrackDiffusion a relevant work?**
>
> Thanks for pointing it out. TrackDiffusion [4] is a training-based method to generate video with given bounding boxes. Therefore, its performance relies heavily on the training data. We have added it as a relevant work.
>
> > **Q8: Quantitative ablations studies would add value of the paper.**
>
> Thanks for your advice. We have added the quantitative ablations in Table 3 of the revised supplementary.
>
> [1] Space-Time Diffusion Features for Zero-Shot Text-Driven Motion Transfer
> [2] Prompt-to-Prompt Image Editing with Cross Attention Control
> [3] MasaCtrl: Tuning-Free Mutual Self-Attention Control for Consistent Image Synthesis and Editing
> [4] TrackDiffusion: Tracklet-Conditioned Video Generation via Diffusion Models

---

> > ### Comment · Reviewer_2SGJ · 2024-11-26
> > **Response from Reviewer 2SGJ**
> >
> > Thank you for addressing my points.
> >
> > - Due to time-limit, I do not expect or require to add this experiments but I believe additionally comparing with LLM-grounded video diffusion's energy optimization method and TrackDiffusion's training solution would add great value.
> > - I believe 10% extra time for the trajectory control is very reasonable and pratical.
> > - For generating 64 frames long video, does the backbone video model naturally support 64frame long video generation capacity? Or is there any other technique for the longer video generation employed?
> >
> > Similar to Reviewer zfvL, I have a concern regarding technical novelty since it's limited to the noise initialization strategy.
> >   However, I appreciate the paper's observations and believe the video gen community need an effective initial noise initialization strategy that is different from inversion-based methods, since the latter has strong constraints on not only trajectory or motion but appearance and structure. In fact, ddim inversion often fails to reflect the motion of the input video even with large steps.
> >
> >   But practically, bounding boxes (groundings) are quite hard and tricky to handle, thus not used as often as spatially aligned conditions like optical flow or depth maps. I believe if the authors somehow later opensource an useful or friendly UI or codes that facilitates FreeTraj's noise initialization strategy that corresponds to user's intention (represented by either text or roughy trajectory input or etc), it would be very helpful to the users and researchers in the community. I raise my score to 6.

---

> > > ### Author Response · Authors · 2024-11-26
> > >
> > > Thank you for considering the additional experimental workload and for recognizing the key highlights of our paper. Once again, we sincerely appreciate the time and effort you have devoted to reviewing our work.

---

> ### Author Response · Authors · 2024-11-28
>
> Sorry for missing one question. For generating 64-frame videos, the backbone video model only naturally supports generating 16-frame videos without quality degradation. Here we use slightly modified FreeNoise [1] to achieve longer video generation. It is a useful plugin and supports various base models, like VideoCrafter, AnimateDiff, and recent CogVideoX. The recent work, FreeLong (Neurips 2024) [2] may have better performance on longer video generation.
>
> [1] FreeNoise: Tuning-Free Longer Video Diffusion via Noise Rescheduling
> [2] FreeLong: Training-Free Long Video Generation with SpectralBlend Temporal Attention

---

> ### Author Response · Authors · 2024-11-28
>
> Sorry. Due to a system issue, duplicate replies were submitted. We have deleted this duplicate reply.

---

### Official Review · Reviewer_zfvL · 2024-10-28

**Soundness:** 3
**Presentation:** 3
**Contribution:** 2
**Rating:** 5
**Confidence:** 5

**Summary:**

The paper presents a tuning-free video diffusion framework for object trajectory control. The authors in this paper first analyze the influence of noise construction for video motion generation and then introduce the FreeTraj. The proposed framework modifies the noise sampling and involves the local trajectory injection in the noise initialization. Besides, the object mask is also integrated into the attention to emphasize the content generation across frames. Experiments on both controllable short- and long-video generation tasks verify the proposed approach.

**Strengths:**

1.	The exploration on the noise initialization for object motion control is interesting and demonstrates the importance of noise structure for content generation in video diffusion models.
2.	Both of the qualitative and quantitative experimental results demonstrate the effectiveness of the proposed approach for tuning-free motion control.
3.	The potential of complex motion control is verified by the model that is combined with FreeNoise under the setting of long video generation.

**Weaknesses:**

1.	My major concern is about the technical contribution which could be limited. The investigation of noise construction is more like an intuitive engineering work and there is no rationale behind. The influence of high-frequency noise part has been also explored in FreeNoise. The trajectory injection seems reasonable but more details should be included. Meanwhile, the authors argue that Peekaboo exploits the hard mask in attention. The motivation of the proposed soft Gaussian mask is not clear either. The hyper-parameter of such Gaussian kernel is not mentioned in the paper.
2.	In the experimental section, the comparison and discussion (line 465 to 470) between FreeTraj and MotionCtrl should be detailed. It is very difficult for readers to judge the technical differences or improvements from these descriptions. Why dose MotionCtrl only roughly control the object trajectory but not align the trajectory? The reason behind these results is not provided or discussed.
3.	One suggestion is about ablation studies. There could be some quantitative analysis for different variants rather than only showing the visual cases. The quantitative evaluation results can reflect the performance from a global view.
4.	Some of the definitions should be aligned. For instance, the $F_{z_T}^{low}$ and $F_{z_T}^{L}$ in Eq. (7).

**Questions:**

Please see the weaknesses.

---

> ### Author Response · Authors · 2024-11-25
>
> Thanks for your valuable comments. We summarize and answer your questions below.
>
> > **Q1: The trajectory injection seems reasonable but more details should be included.**
>
> Thanks for pointing it out. I think you want to say FreeInit (rather than FreeNoise) has explored the influence of high-frequency noise. One of our core technical contributions is exactly trajectory injection, initializing the noise without any DDIM inversion or re-nosing, which is never achieved in FreeInit or other previous work. Please let us know if there is any part unclear of trajectory injection, and we are happy to clarify it. In addition, the motivation for proposing the soft Gaussian mask is that the probability of an object appearing in the bounding boxes is consistent with a Gaussian kernel mostly. In other words, objects may not occupy the boundary areas of the bounding boxes due to their original motion (like waving hands) but always appear in the center of the bounding box. The kernel division in Equation 9 is 3.0, and the kernel shape is the same as the mask shape.
>
> > **Q2: Why does MotionCtrl only roughly control the object trajectory but not align the trajectory?**
>
> Sorry for the confusion. Here “roughly” means that “does not force the object center to align with the trajectory accurately”. In other words, the whole object will follow the trajectory, but the object center does not exactly overlap with the trajectory. This phenomenon may be caused by the training data of MotionCtrl. We clear this statement by removing the word “roughly” in the revised version.
>
> > **Q3: There could be some quantitative analysis for different variants rather than only showing the visual cases.**
>
> Thanks for your advice. We have added the quantitative ablations in Table 3 of the revised supplementary.
>
> > **Q4: Some of the definitions should be aligned.**
>
> Thanks for pointing it out. We have fixed them.
>
> [1] FreeInit: Bridging Initialization Gap in Video Diffusion Models
> [2] FreeNoise: Tuning-Free Longer Video Diffusion via Noise Rescheduling

---

> > ### Comment · Reviewer_zfvL · 2024-11-26
> >
> > Thank you to the authors for their comprehensive feedback and for providing the quantitative ablations of different variants.
> >
> > Nevertheless, I still find it difficult to assess the core contribution of this work. Althought trajectory injection is implicitly implemented through noise initialization, the proposed approach still borrows several techniques from existing works, such as FreeInit and Peekaboo. In particular, the key idea of cross-attention guidance closely resembles the concept introduced in Peekaboo.
> >
> > Moreover, the visual quality of the videos is not very compelling, with noticeable temporal flickering of objects during movement control. Therefore, I will maintain my initial rating for this paper.

---

> > > ### Author Response · Authors · 2024-11-26
> > >
> > > We are glad to solve your previous concerns and thank you for your in-time response thus we can clarify your remaining concerns.
> > >
> > > > **Contributions:**
> > > 1. First, we provide several **inspiring observations and analysis of how initial noise influences the trajectory of generated results**. This is our first core contribution and may inspire the design of future relevant work. Most reviewers should have recognized this contribution.
> > > 2. From the methodology aspect, our method mainly consists of noise guidance and attention guidance. For attention guidance, the main idea of masked attention is inspired by Peekaboo. Our improvement is more like the engineering optimization. Therefore, it is reasonable if you think this part is not novel enough. However, for noise guidance,  we propose the **trajectory injection for the initial noise** based on the observations of how initial noise influences the trajectory of generated results. This is our second core contribution. Only one module of trajectory injection, “high-frequency resampling”, is from FreeInit, and the remaining is our original design. We can easily list two obvious differences: 1) FreeInit never showcases any observation of trajectory control, 2) FreeInit needs to recursively de-noise and re-noise to get a fine initial noise while our FreeTraj directly constructs initial noise. The pipelines of the two papers are totally different. I think the original intention of FreeNoise and FreeInit (or other novel papers) is to see if their work can inspire future work and achieve some new ability beyond their original imagination.
> > >
> > > > **Visual Quality:**
> > > 1. FreeTraj is a tuning-free paradigm and is still limited by the underlying model, such as the consistency of object appearance that easily changes during large movements. We have discussed it in the conclusion. Meanwhile, this also means that the quality of videos generated by Freetraj will improve as the basic model improves.
> > > 2. The noticeable temporal flickering should only appear in the results with complex trajectories (e.g., top-left -> bottom-left -> bottom-right) and long movement. Different from previous trajectory control work, 1. We evaluate FreeTraj on some complex trajectories (e.g., top-left -> bottom-left -> bottom-right) to test the trajectory control capabilities more strictly. However, even for a real video (or video generated by powerful business models), presenting such a long movement within 16 frames will lead to either motion incoherence or motion blur. We find the same motion incoherence on MotionCtrl when given a trajectory with long-range movement. When we use a smaller movement (e.g., mid-left -> mid-right), which is the most common trajectory in previous work, the video quality and motion coherence are well preserved. All generated 16 frames are presented in Figure 5, and no temporal flickering is observed. The temporal flickering problem should be solved naturally as the base model is improved (generating more frames and stronger consistency).
> > >
> > > We hope this explanation provides clarity for your remaining concerns. We fully understand that the definition of novelty is somehow subjective. Therefore, we respect your final decision. Once more, we appreciate the time and effort you've dedicated to our paper.

---

> > > ### Author Response · Authors · 2024-12-01
> > > **Follow-up Reply**
> > >
> > > Sorry to bother you. We make a detailed **Contribution Clarification** in the new common response and list the differences with related papers. Our core contributions include **Observations and analysis of the relations between initial noise and trajectory** and proposed **Trajectory Injection** based on these observations. Since all motivation, design logic, and details are carefully introduced in our observations and analysis part (Section 3.2), we streamlined the presentation in Section 3.3.1 by focusing on the final operation, avoiding repetitive and overly detailed instructions. This may have inadvertently led to the novelty of this part being overlooked. We sincerely apologize for any confusion caused and respectfully invite reviewers to assess the design logic and the final noise injection solution as an integrated whole when evaluating the article's novelty.
> > >
> > > If you have further questions regarding the points outlined in the Contribution Clarification, please do not hesitate to reach out. We appreciate your time and consideration.

---

### Official Review · Reviewer_sSw8 · 2024-11-02

**Soundness:** 3
**Presentation:** 3
**Contribution:** 2
**Rating:** 5
**Confidence:** 5

**Summary:**

This paper proposes a diffusion-based video generation method for controlling the trajectory of moving objects in a zero-shot manner and investigates the impact of initial noise on the trajectories of moving objects. By guiding the generated target in noise and attention, the proposed method can generate controlled trajectory of the target. Experimental results demonstrate the effectiveness of this method.

**Strengths:**

- The method described in this paper does not require any training, which can significantly reduce computational overhead.
- This method analyzes the high and low frequencies of the initial noise and Attention Isolation, and proposes corresponding utilization methods and solutions.
- The provided generated video looks good in controlling.

**Weaknesses:**

- In the paper, the author uses a box to control the motion of objects, but in the experiments, the objects are not actually inside the box; they merely maintain a trajectory consistent with that of the box.
- The method performs significantly worse than the Direct approach on the FVD and KVD metrics, and the author does not analyze this. This could potentially have negative effects on the clarity of the generated videos and on maintaining the identity of the objects.
- This method shows some weaknesses in generating complex and random trajectories. However, it seems feasible to achieve better results by increasing the control strength, according to the method. The author does not further analyze this aspect. The degree of control could potentially impact video quality. It would be beneficial for the author to include an ablation study on the intensity of control to better understand its effects.

**Questions:**

- The author mentions that this method can be used for LLM trajectory planners, but how this can be implemented? This lack of detail could leave readers uncertain about the practical application of the method in such contexts.
- Since the method does not require training, it could be applicable to any diffusion-based video generation method, such as Stable Video Diffusion or OpenSora. Could you provide more experiments with different video-diffusion baselines? This universality could potentially make it a versatile tool in the field of video synthesis, especially for applications requiring rapid deployment without the need for extensive training data.
- As in zero-shot setting, Motion-Zero[1] has some similar features, modifying initial noise, and guidance in attention. in video generation. What's the difference between the proposed method and [1]?

[1] C. Chen et al. Motion-Zero: Zero-Shot Moving Object Control Framework for Diffusion-Based Video Generation. Arxiv 2024.

---

> ### Author Response · Authors · 2024-11-25
>
> Thanks for your valuable comments. We summarize and answer your questions below.
>
> > **Q1: The objects are not actually inside the box.**
>
> Thanks for pointing it out. We only use bounding boxes as the guidance signals for trajectory control. Due to the strong prior of VideoCrafter2, we only accurately control the trajectory but roughly control the size of objects. Specificity, VideoCrafter2 tends to generate results fitting the distribution learned from the training data. For example, as shown in Figure 6, when the bear climbs down, it will follow the tree. As shown in Figure 12,  if we add the control intensity and force the size of generated objects to break the learned prior, the quality may be hurt.
>
> > **Q2: The method performs significantly worse than the Direct approach on the FVD and KVD metrics.**
>
> Thanks for pointing it out. FVD and KVD evaluate video quality by measuring the distribution distances of two datasets. We claim that large motion in generated videos will make the distribution deviate from direct sampling, which only contains small movements. The dynamic degree of direct generation is 0.554, which is close to reference videos, while the dynamic degree of FreeTraj is 0.982. The large dynamics gap causes worse FVD and KVD. To prove it, we also have added Ours-SmallMove, the same setting but with the new trajectories of small movements. When the movement decreases, the FVD and KVD are significantly improved.
>
> > **Q3: It would be beneficial for the author to include an ablation study on the intensity of control to better understand its effects.**
>
> Thanks for your advice, we have added it in Table 3 and Figure 13. In this paper, we select a restrained control intensity by default to guarantee that most generated cases do not contain artifacts. However, users can get a better traject-controllable result by adding the control intensity and sampling more times with different random seeds to get the results without artifacts.
>
> > **Q4: How to use LLM trajectory planners for this method?**
>
> Thanks for pointing it out. We slightly modified the prompt from the paper, LLM-Grounded Video Diffusion Models, and have showcased the results in Figure 14 of the revised version.
>
> > **Q5: Could you provide more experiments with different video-diffusion baselines?**
>
> Thanks for your advice. In the original submission, Figure 10 in the original submission exhibits that objects and textures in the video generated by AnimateDiff also flow in the same direction as the initial noises. According to your advice, we try the whole FreeTraj on AnimateDiff. As shown in Figure 13 of the revised version, FreeTraj successfully controls the trajectory in AnimateDiff. We will try other DiT-based models in the future like OpenSora. Currently, pure FreeTraj only supports text-to-video generation thus not applicable for image-to-video model, Stable Video Diffusion. However, recent work SG-I2V [1] has explored applying similar techniques to image-to-video models.
>
> > **Q6: What's the difference between the proposed method and Motion-Zero?**
>
> Thanks for pointing it out. The key differences between FreeTraj and Motion-Zero [2] lie in two main aspects:
>
> 1. Motion-Zero is an optimization-based approach that requires iterative updates to the initialized noise to ensure that the generated object remains centered within the given bounding boxes. In contrast, FreeTraj is a tuning-free method, achieving object control through resampled initialized noise and a meticulously designed framework.
>
> 2. Motion-Zero is limited to replicating object motions derived from existing videos. It attains the original initialized noise by reversing the given video using DDIM, a process that is both time-consuming and constrained in terms of control diversity. In comparison, FreeTraj leverages a fast Fourier transformation to resample noise efficiently and supports diverse, user-defined moving boxes, enabling more versatile and efficient object motion control.
>
> We have added it as a relevant work.
>
> [1] SG-I2V: Self-Guided Trajectory Control in Image-to-Video Generation
> [2] Motion-Zero: Zero-Shot Moving Object Control Framework for Diffusion-Based Video Generation

---

> > ### Comment · Reviewer_sSw8 · 2024-11-26
> >
> > Thanks for the authors' detailed feedback on my questions and concerns.
> >
> > However, I quite agree with Reviewer zfvL and am still concerned about the novelty and the significance of the current version of the work. The key idea is similar to several existing works, i.e. FreeInit, Peekaboo and Motion-zero. And, the generated video has inferior performance in terms of FVD and KVD metrics.
> >
> > Therefore, I will maintain my initial rating for this paper.

---

> > > ### Author Response · Authors · 2024-11-26
> > >
> > > We are glad to solve your previous concerns and thank you for your in-time response thus we can clarify your remaining concerns.
> > >
> > > > **Contributions:**
> > > 1. First, we provide several inspiring **observations and analysis of how initial noise influences the trajectory of generated results**. This is our first core contribution and may inspire the design of future relevant work. Most reviewers should have recognized this contribution.
> > > 2. From the methodology aspect, our method mainly consists of noise guidance and attention guidance. For attention guidance, the main idea of masked attention is inspired by Peekaboo. Our improvement is more like the engineering optimization. Therefore, it is reasonable if you think this part is not novel enough. However, for noise guidance,  we propose the **trajectory injection for the initial noise** based on the observations of how initial noise influences the trajectory of generated results. This is our second core contribution. Only one module of trajectory injection, “high-frequency resampling”, is from FreeInit, and the remaining is our original design. We can easily list two obvious differences: 1) FreeInit never showcases any observation of trajectory control, 2) FreeInit needs to recursively de-noise and re-noise to get a fine initial noise while our FreeTraj directly constructs initial noise. The pipelines of the two papers are totally different. **Motion-Zero needs the trajectory from the given video and the additional DDIM inversion. We have listed the difference between FreeTraj and Motion-Out in detail when the first response.** Please indicate what you disagree with. I think the original intention of FreeNoise and FreeInit (or other novel papers) is to see if their work can inspire future work and achieve some new ability beyond their original imagination.
> > >
> > > > **Visual Quality:**
> > > 1. FreeTraj is a tuning-free paradigm and is still limited by the underlying model, such as the consistency of object appearance that easily changes during large movements. We have discussed it in the conclusion. Meanwhile, this also means that the quality of videos generated by Freetraj will improve as the basic model improves.
> > > 2. The noticeable temporal flickering should only appear in the results with complex trajectories (e.g., top-left -> bottom-left -> bottom-right) and long movement. Different from previous trajectory control work, 1. We evaluate FreeTraj on some complex trajectories (e.g., top-left -> bottom-left -> bottom-right) to test the trajectory control capabilities more strictly. However, even for a real video (or video generated by powerful business models), presenting such a long movement within 16 frames will lead to either motion incoherence or motion blur. We find the same motion incoherence on MotionCtrl when given a trajectory with long-range movement. When we use a smaller movement (e.g., mid-left -> mid-right), which is the most common trajectory in previous work, the video quality and motion coherence are well preserved. All generated 16 frames are presented in Figure 5, and no temporal flickering is observed. The temporal flickering problem should be solved naturally as the base model is improved (generating more frames and stronger consistency).
> > >
> > > > **FVD and KVD:**
> > > 1. In our first response, we carefully give the reason why FVD and KVD will be larger in the trajectory control task. **There is a large gap in the dynamics degree between evaluated videos and reference videos, causing worse FVD and KVD.**
> > > 2. In addition, the FVD reported in MotionCtrl (https://arxiv.org/pdf/2312.03641) is around 1000 or even higher, which still does not hurt its value on motion control.
> > >
> > > We hope this explanation provides clarity for your remaining concerns. **We really think you want to help us improve this paper and have put much effort into implementing all the experiments you advised, i.e., quantitative ablation, LLM-planed generation, and FreeTraj + AnimateDiff.** Please do not ignore them. But still, we fully understand that the definition of novelty is somehow subjective. Therefore, we respect your final decision. Once more, we appreciate the time and effort you've dedicated to our paper.

---

> > > ### Author Response · Authors · 2024-12-01
> > > **Follow-up Reply**
> > >
> > > Sorry to bother you. We make a detailed **Contribution Clarification** in the new common response and list the differences with related papers. Our core contributions include **Observations and analysis of the relations between initial noise and trajectory** and proposed **Trajectory Injection** based on these observations. Since all motivation, design logic, and details are carefully introduced in our observations and analysis part (Section 3.2), we streamlined the presentation in Section 3.3.1 by focusing on the final operation, avoiding repetitive and overly detailed instructions. This may have inadvertently led to the novelty of this part being overlooked. We sincerely apologize for any confusion caused and respectfully invite reviewers to assess the design logic and the final noise injection solution as an integrated whole when evaluating the article's novelty.
> > >
> > > If you have further questions regarding the points outlined in the Contribution Clarification, please do not hesitate to reach out. We appreciate your time and consideration.

---

### Official Review · Reviewer_3hqH · 2024-11-05

**Soundness:** 2
**Presentation:** 3
**Contribution:** 2
**Rating:** 6
**Confidence:** 4

**Summary:**

This paper analyze the trajectory-controllable video generation and introduce Free_Traj, a training-free motion control method.

**Strengths:**

- It is intuitive and reasonable to use noise and motion guidance.

- Noise resampling is an interesting idea.

- The analysis is comprehensive, and the results seem good to me.

**Weaknesses:**

- Is mIoU a good metric for trajectory control?


- Can this method unify trajectory control and motion control? For example, a man waves his hand. In terms of motion, it is a "wave" action; regarding trajectory, the hand follows its specific path. Can we unify them to achieve more realistic motion control?

**Questions:**

Please see the weakness.

---

> ### Author Response · Authors · 2024-11-25
>
> Thanks for your valuable comments. We summarize and answer your questions below.
>
> >  **Q1: Is mIoU a good metric for trajectory control?**
>
> Thanks for pointing it out. Compared to mIoU, Centroid Distance is better for measuring trajectory control. However, in this paper, we use bounding boxes as the guidance signals. Therefore, we evaluate both metrics for a more comprehensive reference. If you think other metrics can help evaluate the trajectory control, we are happy to test them.
>
> > **Q2: Can this method unify trajectory control and motion control?**
>
> Thanks for your advice. It is a good application. Currently, FreeTraj can not achieve unified control directly. This is why we refrained from naming the task trajectory control rather than motion control. In the future, combining some local control technologies from DragDiffusion [1] and DragAnything [2] may potentially solve this challenging unified task.
>
> [1] DragDiffusion: Harnessing Diffusion Models for Interactive Point-based Image Editing
> [2] DragAnything: Motion Control for Anything using Entity Representation

---

> > ### Comment · Reviewer_3hqH · 2024-11-25
> >
> > Thank you for your response. I think FreeTraj  is a good  trajectory control method. After considering your comment and other reviews, I have decided to maintain my current rating.

---

> > > ### Author Response · Authors · 2024-11-25
> > >
> > > Thanks for your fair evaluation! Once more, we appreciate the time and effort you've dedicated to our paper.

---

### Author Response · Authors · 2024-11-25
**Common Response**

We sincerely thank all reviewers for their constructive suggestions and recognition of our work. We are encouraged that reviewers find that our observation and analysis are **“convincing and well-structured.”** (Reviewer 3hqH, sSw8, zfvL, 2SGJ); our proposed method achieves **good traject control** (Reviewer 3hqH, sSw8, zfvL, 2SGJ); and tuning-free strategy is **valuable and meaningful** (Reviewer sSw8, 2SGJ). We have separate responses for each reviewer and also updated our submission to include the following changes according to reviewers' feedback. Note that the main revisions in the main paper and appendix are highlighted in blue:

1. We have added several missing relevant papers in the Related Work.
2. According to the advice from reviewers, we have added four parts to the Appendix: Quantitative Ablation (Section C.3), Control Intensity (Section C.4), Method Compatibility (Section C.5), and LLM-Planed Generation (Section C.6).

Please do not hesitate to let us know if you have any additional comments or there are more clarifications that we can offer.

---

### Author Response · Authors · 2024-11-30
**Contribution Clarification**

Thanks to the constructive suggestions and active discussions from the reviewers, we have addressed most reviewers’ concerns through sufficient additional experiments and detailed explanations. The remaining concern is mainly about the contributions thus we make this clarification.

>**Our core contributions include:**

**1. Observations and analysis of the relations between initial noise and trajectory.** We first propose the **Noise Flow**, a naive way to prove that initial noise can guide the trajectory of generated videos. However, this rude manipulation will cause obvious artifacts. Therefore, we use **High-Frequency Noise Resampling** to preserve the video quality while still maintaining the ability of trajectory control. Finally, we inject the noise locally to separate the control of foreground and background.

**2. Trajectory Injection.** Based on our observations and analysis, we propose trajectory injection, which **directly** injects the target trajectory into the low-frequency part of the initial noise. The difficulties are how to construct a trajectory directly **without any iterative process** and how to preserve the quality after manipulating the initial noise.

Since all motivation, design logic, and details are carefully introduced in our observations and analysis part (Section 3.2), we have omitted the repetitive and complicated instructions and only simply introduced the final operation in Section 3.3.1. That should be the reason why the novelty of this part is ignored. We apologize for the confusion and invite reviewers to consider the design logic and final solution of noise injection as a whole when evaluating the novelty of the article. We sincerely thank reviewer 2SGJ for recognizing that our effective initial noise initialization strategy will bring enlightenment value to the video gen community.

Additionally, our other two contributions consist of (1) improving the masked attention based on previous work [2,3,4], and
(2) extending the control mechanism to achieve longer and larger video generation with a controllable trajectory. These two contributions are effective but belong to engineering improvement and technical application. We agree that they are not very novel and the novelty discussion can focus on our core contributions.

>**Difference with FreeInit [1]:**

FreeTraj only utilizes the idea of **High-Frequency Noise Resampling** from FreeInit and the two methods are obviously different in terms of motivation, pipeline, and dynamics:
1. **Motivation.** FreeInit targets to improve the generated quality by reducing the gap between training and inference. FreeTraj targets to control the trajectory of generated videos.
2. **Pipeline.** FreeInit needs to find a better initial noise through an iterative process, which is time-consuming. However, FreeTraj directly constructs the target initial noise without any iterative process.
3. **Dynamics.** Compared to the direct inference of the base model, videos generated by FreeInit tend to have fewer dynamics, while videos generated by FreeTraj have significantly larger dynamics.

>**Difference with Motion-Zero [5]:**

Motion-Zero is limited to replicating object motions derived from existing videos. It attains the original initialized noise by **reversing the given video using DDIM**, a process that is both time-consuming and constrained in terms of control diversity. In comparison, FreeTraj **directly** injects the target trajectory into the low-frequency part of the initial noise, enabling more versatile and efficient object motion control. In other words, Motion-Zero realizes that initial noise can guide the trajectory of generated videos but has no idea about constructing initial noise directly with preserved video quality and trajectory control ability.

[1] FreeInit: Bridging Initialization Gap in Video Diffusion Models
[2] Peekaboo: Interactive video generation via masked-diffusion
[3] Trailblazer: Trajectory control for diffusion-based video generation
[4] Direct-a-Video: Customized Video Generation with User-Directed Camera Movement and Object Motion
[5] Motion-Zero: Zero-Shot Moving Object Control Framework for Diffusion-Based Video Generation

---

### Meta-Review · Area_Chair_qbHK · 2024-12-19

**Metareview:**

The submission propose a training-free approach for object-trajectory-controllable in video generation. Specifically, the authors (1) analyze how the initial noise affects generated videos including the motion and output quality, (2) propose to adjust the initial noise and noise resampling for high frequency signals, and (3) introduce a soft-masking mechanism based on object locations.

Strength
* The idea of directly manipulating initial noise to influence object trajectories is interesting, intuitive, and requires no additional training.
* The paper offers a comprehensive analysis of how noise and attention contribute to video generation.
* The proposed method can produce promising results in terms of controlling object trajectories in generated videos.

Weakness
* The chosen metrics for trajectory control evaluation are not fully justified.
* The controllability is not accurate. Objects may not precisely follow the inpu bounding boxes.
* The generated videos show much worse quality metrics (FVD, KVD). The authors’ rebuttal attributes this to large motion within the video, but there's no analysis for how control may affect video quality and the implication of this observation.
* The technical contribution is limited. The proposed noise and attention manipulations has been studied in the same context.
* The paper lacks thorough analysis and detailed comparisons with closely related baseline methods, leaving the novelty and advantages of the approach unclear.
* While the authors mention using an LLM-based trajectory planner, the claim lacks justification.

The reviewers raised concerns about the technical novelty of the submission, as the proposed noise and attention manipulations have been previously explored in similar contexts. The lack of detailed analysis and comprehensive comparisons with relevant baseline methods further diminishes the clarity and significance of the contributions. Additionally, there are concerns about the output quality, including inaccuracies in trajectory control and poor performance on video quality metrics such as FVD and KVD. While the authors attributed the lower metrics to large motion dynamics in the videos, this explanation does not sufficiently address broader concerns regarding the practical effectiveness of the proposed method in generating high-quality, controllable video outputs.

**Additional Comments On Reviewer Discussion:**

While the rebuttal addressed the clarification questions raised by the reviewers, concerns regarding the technical contributions and output quality persisted even after the rebuttal and discussion.

---

### Decision · Program_Chairs · 2025-01-22

Reject